# A Zero-and-One Inflated Cosine Geometric Distribution and Its Application

Sunisa Junnumtuam [†], Sa-Aat Niwitpong *,[†] and Suparat Niwitpong [†]

Department of Applied Statistics, Faculty of Applied Science, King Mongkut's University of Technology North Bangkok, Bangkok 10800, Thailand
* Correspondence: sa-aat.n@sci.kmutnb.ac.th
† These authors contributed equally to this work.

**Abstract:** Count data containing both excess zeros and ones occur in many fields, and the zero-and-one inflated distribution is suitable for analyzing them. Herein, we construct confidence intervals (CIs) for the parameters of the zero-and-one inflated cosine geometric (ZOICG) distribution constructed by using five methods: a Wald CI based on the maximum likelihood estimate, equal-tailed Bayesian CIs based on the uniform or Jeffreys prior, and the highest posterior density intervals based on the uniform or Jeffreys prior. Their efficiencies were compared in terms of their coverage probabilities and average lengths via a simulation study. The results show that the highest posterior density intervals based on the uniform prior performed the best in most cases. The number of new daily COVID-19-related deaths in Luxembourg in 2020 involving data with a high proportion of zeros and ones were analyzed. It was found that the ZOICG model was appropriate for this scenario.

**Keywords:** zero-and-one inflated cosine geometric distribution; Bayesian estimation; Metropolis–Hasting algorithm; confidence intervals

**MSC:** 62F25





## 1. Introduction

Overdispersed count data with excess zeros frequently occur in various fields such as natural science (e.g., the number of torrential rainfall incidences at the Daegu and Busan rain gauge stations in South Korea [1]), medical science (e.g., the DMFT (decayed, missing, and filled teeth) index in dentistry [2], the number of falls by people with Parkinson's disease [3], and the number of daily COVID-19 deaths in Thailand [4]), and insurance (e.g., the frequency of health insurance claims [5]). Although the Poisson model is widely used to analyze such discrete data, the strong assumption of the equality of the mean and variance implies that it is inadequate for modeling data where the variance is larger than the mean, which is termed "overdispersion" [6] and can arise in various ways.

One common source of overdispersion is when the observed counts contain excess zeros, which has motivated the creation of modified count models such as the zero-inflated (ZI) and hurdle models. These two classes can be considered finite mixture models of two subpopulations (components): (1) the observations contain zeros with a probability $\omega$, and (2) the observations occur with a probability $1 - \omega$ from the baseline distribution of a ZI model and the zero-truncated probability mass function (pmf) (the nonzero part) of a hurdle model. One of the most popular ZI models is the ZI Poisson (ZIP) model, where the Poisson distribution is the baseline distribution [7].

However, overdispersion arises when large numbers of zeros and ones occur simultaneously. Since the ZI model is not an appropriate for model such data, it has been extended to the zero-and-one-inflated (ZOI) model combined with several other distributions to produce the ZOI Poisson (ZOIP), ZOI geometric, ZOI negative binomial-beta exponential

(ZOINB-BE), and ZOIP-Lindley mixed distributions. Zhang et al. [8] explored the ZOIP distribution extended from the ZIP distribution [9]. Five equivalent stochastic representations for a ZOIP random variable were presented, and their important distributional properties were derived. Tang et al. [10] provided a ZOIP model and analyzed two real datasets of Legionnaires' disease cases in Singapore and accidental deaths in Detroit, USA, both of which contained high proportions of zeros and ones. They used the data augmentation method to obtain the maximum likelihood estimators (MLEs) via the EM algorithm and Bayesian estimations via Gibbs sampling. By obtaining the lowest Akaike information criterion (AIC), deviance information criterion (DIC), and widely applicable information criterion (WAIC) values for the ZOIP, they reported that it was more appropriate for these datasets than the ZIP model. Liu et al. [11] analyzed two real datasets of Legionnaires' disease cases in Singapore and accidental deaths in Detroit, USA by using the ZOIP model with both the maximum likelihood and Bayesian estimation. They found that the ZOIP model was more appropriate than the ZIP model for analyzing these two real datasets. Liu et al. [12] studied the number of daily accidental deaths in 1994, which were available from the NMMAPS database, by using a ZOIP regression model and investigated the maximum likelihood and Bayesian estimation, the expectation maximization algorithm, the generalized expectation maximization algorithm, and Gibbs' sampling to estimate its parameters. They found that the ZOIP regression model was more appropriate than the ZIP model for analyzing this accidental deaths dataset, as it attained lower AIC, Bayesian information criterion (BIC), DIC, and WAIC values. Xiao et al. [13] constructed a ZOI geometric distribution regression model and introduced Pólya-gamma latent variables in the Bayesian inference, and they found that it was more suitable for analyzing a doctoral dissertation dataset than the ZOIP regression model. Jornsatian and Bodhisuwan [14] presented a ZOI negative binomial-beta exponential (ZOINB-BE) distribution, investigated some of its important properties, and analyzed three real datasets: the number of visits to the doctor in Germany in 1998 (the COUNT package in the R programming suite), the number of accidental injuries in the US in 2001 [15,16], and the number of monthly crimes in Greece from 1982 to 1993 [17,18]. They found that the ZOINB-BE distribution was most appropriate to fit these data by attaining the lowest log-likelihood, AIC, mean absolute error, and root mean squared error values. Tajuddin et al. [19] introduced the ZOIP–Lindley distribution and developed MLEs and method-of-moment estimators for its parameters. They found that it was the most appropriate for analyzing two datasets: the number of criminal acts [20] and the number of stillbirths in New Zealand white rabbits [21]. Mohammadi et al. [22] introduced a zero-and-one inflated INAR(1) process with a Poisson–Lindley distribution and analyzed the number of abortions of animals reported monthly, which contained large proportions of zeros and ones. They found that the proposed model had the best fit based on the AIC, BIC, log-likelihood, and root mean square differences between the observation and prediction (RMS) criteria.

Aside from these mixed distributions, another one is the two-parameter discrete cosine geometric (CG) distribution, which was proposed by Chesneau [23]. This belongs to the family of weighted geometric distributions, with its pmf given by

$$P(X = k \mid p, \theta) = C_{p,\theta} p^k [\cos(k\theta)]^2, k \in \mathbb{N}, \tag{1}$$

where $\theta \in [0, \frac{\pi}{2}]$

$$C_{p,\theta} = \frac{2(1-p)(1 - 2p\cos(2\theta) + p^2)}{2 + p((p-3)\cos(2\theta) + p - 1)}. \tag{2}$$

If $\theta = 0$, then we can obtain $C_{p,\theta} = 1 - p$, and $X$ is a standard geometric distribution. A weighted geometric distribution makes the CG distribution more flexible than the standard geometric distribution. The former is better for analyzing overdispersed data than the Poisson, geometric, negative binomial (NB), and weighted NB distributions. Junnumtuam et al. [24] extended it and proposed the ZICG distribution (with CG as the baseline distribution), and they reported that it is appropriate for fitting overdispersed

count data containing excess zeros, such as the number of daily positive COVID-19 cases at the Tokyo 2020 Olympic Games.

Since studies on ZOI count data have gained much research interest, and the CG distribution is appropriate for overdispersed data, we can extend it to the novel, four-parameter, discrete ZOICG distribution and derive some of its statistical properties such as the pmf, moment-generating function (mgf), mean, variance, and Fisher information. Moreover, its parameters are estimated by deriving their confidence intervals (CIs). There are several examples of applying CIs to analyze ZI and ZOI count data. Liu et al. [11] considered both point and interval estimation for the parameters of a ZOIP model and compared Bayesian estimation using either the Jeffreys or reference prior with the MLE method via Monte Carlo simulation. The results indicated that the Bayesian estimates performed slightly better when the sample size was small or moderate. Tian et el. [25] proposed CIs for the mean of a zero-and-one inflated population by using the jackknife empirical likelihood and adjusted jackknife empirical likelihood methods. Wald CIs were constructed for the parameters in the Bernoulli component of the ZIP and hurdle models in [26] and for the ZIP mean in [27].

This motivated us to study the confidence intervals for the parameters of the ZOICG distribution to estimate the overdispersed data, which contain a large proportion of zeros and ones with a high index of dispersion, by using five methods: a Wald CI based on the MLE, equal-tailed Bayesian CIs based on the uniform or Jeffreys prior, and highest posterior density (HPD) intervals based on the uniform or Jeffreys prior. Furthermore, real data containing excess zeros and ones (the number of new daily COVID-19 deaths in Luxembourg in 2020) were used to investigate their efficacies.

## 2. The ZOICG Distribution

A random variable $Y$ in a ZOICG model can be derived as follows:

$$Y = (1 + Z)\eta + ZX, \tag{3}$$

where $Z$ is a Bernoulli random variable with a success probability $(1 - (\omega_1 + \omega_2))$, $\eta$ is a Bernoulli random variable with a success probability $\omega_2/(\omega_1 + \omega_2)$, $X$ follows a CG distribution with parameters $p$ and $\theta$, and $Z, \eta$, and $X$ are mutually independent. Hence, the relationship between $Y$ and $(Z, \eta, X)$ becomes

$$Y = (1 - Z)\eta + ZX = \begin{cases} \eta, & \text{with probability} \quad \omega_1 + \omega_2 \\ X, & \text{with probability} \quad 1 - (\omega_1 + \omega_2) \end{cases}, \tag{4}$$

which follows a ZOICG distribution with parameters $\omega_1, \omega_2, p$, and $\theta$. This can be verified by

$$
\begin{aligned}
Pr(Y = 0) =& (\omega_1 + \omega_2)(1 - (\frac{\omega_2}{(\omega_1 + \omega_2)})) + (1 - (\omega_1 + \omega_2))P(X = 0 \mid p, \theta) \\
=& \omega_1 + (1 - \omega 1 - \omega_2)C_{p,\theta}, \\
Pr(Y = 1) =& (\omega_1 + \omega_2)(\frac{\omega_2}{(\omega_1 + \omega_2)})) + (1 - (\omega_1 + \omega_2))P(X = 1 \mid p, \theta) \\
=& \omega_2 + (1 - \omega 1 - \omega_2)C_{p,\theta}p(\cos\theta)^2, \\
Pr(Y = k) =& (1 - \omega 1 - \omega_2)P(X = k \mid p, \theta); k \geq 2 \\
=& (1 - \omega 1 - \omega_2)C_{p,\theta}p^k(\cos(k\theta))^2.
\end{aligned}
\tag{5}
$$

Hence, the pmf of the nonnegative integer-valued random variable $Y$ is

$$P(Y = k \mid \omega_1, \omega_2, p, \theta) = \begin{cases} \omega_1 + (1 - \omega_1 - \omega_2)C_{p,\theta} & ; k = 0 \\ \omega_2 + (1 - \omega 1 - \omega_2)C_{p,\theta}p(\cos\theta)^2 & ; k = 1, \\ (1 - \omega 1 - \omega_2)C_{p,\theta}p^k(\cos(k\theta))^2 & ; k \geq 2 \end{cases} \tag{6}$$

where $\omega_1, \omega_2 \in (0,1), p \in (0,1)$, and $\theta \in [0, \frac{\pi}{2}]$.

### 2.1. Moment-Generating Function

The moment-generating function of $Y \sim ZOICG(\omega_1, \omega_2, p, \theta)$ can be derived by using $E(W_1) = E[E(W_1 \mid W_2)]$ [8] as follows:

$$
\begin{aligned}
M_Y(t) =& E[\exp(tY)] \\
=& E\{\exp[t(1-Z)\eta + tZX]\} \\
=& E\{E\{\exp[t(1-Z)\eta + tZX] \mid Z\}\} \\
=& E[M_\eta(t(1-Z)) \cdot M_X(tZ)] \\
=& E\left[[1 - \phi + \phi\exp(t(1-Z))] \cdot \frac{C_{p,\theta}}{2}\left[\frac{1}{1 - pe^{tZ}} + \frac{1 - pe^{tZ}\cos(2\theta)}{1 - 2pe^{tZ}\cos(2\theta) + (pe^{tZ})^2}\right]\right] \\
=& \psi(1 - \phi + \phi\exp(t)) + (1-\psi)\frac{C_{p,\theta}}{2}\left[\frac{1}{1 - pe^t} + \frac{1 - pe^t\cos(2\theta)}{1 - 2pe^t\cos(2\theta) + (pe^t)^2}\right],
\end{aligned}
\tag{7}
$$

$$
M_Y'(t) = \psi\phi e^t + (1-\psi)\frac{C_{p,\theta}}{2}pe^t\left[\frac{1}{(1-pe^t)^2} + \frac{\cos(2\theta)(p^2e^{2t}+1) - 2pe^t}{(1 - 2pe^t\cos(2\theta) + p^2e^{2t})^2}\right],
\tag{8}
$$

$$
M_Y'(t=0) = E(Y) = \psi\phi + (1-\psi)\frac{C_{p,\theta}}{2}p\left[\frac{1}{(1-p)^2} + \frac{\cos(2\theta)(p^2+1) - 2p}{(1 - 2p\cos(2\theta) + p^2)^2}\right],
\tag{9}
$$

$$
M_Y''(t) = \psi\phi e^t + (1-\psi)\frac{C_{p,\theta}}{2}pe^t\left[\frac{pe^t+1}{(1-pe^t)^3} + \frac{\cos(2\theta)(p^4e^{4t}-1) + p^3e^{3t}(\cos(4\theta)-3) + pe^t(3 - \cos(4\theta))}{(1 - 2pe^t\cos(2\theta) + p^2e^{2t})^3}\right],
\tag{10}
$$

$$
\begin{aligned}
M_Y''(t=0) =& E(Y^2) \\
=& \psi\phi + (1-\psi)\frac{C_{p,\theta}}{2}p\left[\frac{p+1}{(1-p)^3} + \frac{\cos(2\theta)(p^4-1) + p^3(\cos(4\theta)-3) + p(3 - \cos(4\theta))}{(1 - 2p\cos(2\theta) + p^2)^3}\right],
\end{aligned}
\tag{11}
$$

where $\phi = \omega_2/(\omega_1 + \omega_2)$ and $\psi = \omega_1 + \omega_2$. Thus

$$
E(Y) = \omega_2 + (1 - (\omega_1 + \omega_2))\frac{C_{p,\theta}}{2}p\left[\frac{1}{(1-p)^2} + \frac{\cos(2\theta)(p^2+1) - 2p}{(1 - 2p\cos(2\theta) + p^2)^2}\right],
\tag{12}
$$

$$
E(Y^2) = \omega_2 + (1 - (\omega_1 + \omega_2))\frac{C_{p,\theta}}{2}p\left[\frac{p+1}{(1-p)^3} + \frac{\cos(2\theta)(p^4-1) + p^3(\cos(4\theta)-3) + p(3 - \cos(4\theta))}{(1 - 2p\cos(2\theta) + p^2)^3}\right].
\tag{13}
$$

### 2.2. Maximum Likelihood Estimation

Alternatively, we can develop the second form of the ZOICG model by using the following transformation of the first form with a similar method to that in [11]. The respective probabilities of $Y$ being zeros and ones are

$$
\begin{aligned}
q_0 =& \omega_1 + (1 - \omega_1 - \omega_2)C_{p,\theta} \\
q_1 =& \omega_2 + (1 - \omega_1 - \omega_2)C_{p,\theta}p(\cos\theta)^2.
\end{aligned}
\tag{14}
$$

Hence, the pmf from Equation (6) becomes

$$P(Y = k \mid q_0, q_1, p, \theta) = \begin{cases} q_0 & ; k = 0 \\ q_1 & ; k = 1, \\ \frac{1 - q_0 - q_1}{1 - C_{p,\theta} - C_{p,\theta} p (\cos \theta)^2} C_{p,\theta} p^k (\cos(k\theta))^2 & ; k \geq 2 \end{cases} \tag{15}$$

where $q_0 \geq 0, q_1 \geq 0, q_0 + q_1 \leq 1, p \in (0, 1)$, and $\theta \in [0, \frac{\pi}{2}]$.

Based on a random sample $\mathbf{Y} = (Y_1, ..., Y_n)$ of a size $n$ from the ZOICG model in Equation (15), we define the likelihood function $L = L(q_0, q_1, p, \theta, \mid Y)$ as follows:

$$L(q_0, q_1, p, \theta, \mid \mathbf{Y}) = \frac{q_0^{S_0} q_1^{S_1} (1 - q_0 - q_1)^{n - S_0 - S_1} C_{p,\theta}^{n - S_0 - S_1} p^S \prod_{Y_i \geq 2} (\cos(Y_i \theta))^2}{(1 - C_{p,\theta} - C_{p,\theta} p (\cos \theta)^2)^{n - S_0 - S_1}}, \tag{16}$$

where $S_0 = S_0(\mathbf{Y}) = \sum_{i=1}^n I(Y_i = 0), S_1 = S_1(\mathbf{Y}) = \sum_{i=1}^n I(Y_i = 1)$, and $S = S(\mathbf{Y}) = \sum_{Y_i \geq 2} Y_i$. The mean and variance of $Y$ are respectively given by

$$\begin{aligned} E(Y) = & q_1 + \left( \frac{1 - q_0 - q_1}{1 - C_{p,\theta} - C_{p,\theta} p (\cos \theta)^2} \right) \times \\ & \left( p \frac{C_{p,\theta}}{2} \left[ \frac{1}{(1 - p)^2} + \frac{\cos(2\theta)(1 + p^2) - 2p}{(1 - 2p \cos(2\theta) + p^2)^2} \right] - C_{p,\theta} p (\cos \theta)^2 \right), \end{aligned} \tag{17}$$

$$\begin{aligned} V(Y) = & q_1 + \left( \frac{1 - q_0 - q_1}{1 - C_{p,\theta} - C_{p,\theta} p (\cos \theta)^2} \right) \times \\ & \left( p \frac{C_{p,\theta}}{2} \left[ \frac{1 + p}{(1 - p)^3} - \frac{\cos(2\theta)(p^4 - 1) + p^3 (\cos(4\theta - 3)) + p(3 - \cos(4\theta))}{(1 - 2p \cos(2\theta) + p^2)^3} - C_{p,\theta} p (\cos \theta)^2 \right] \right) \\ & - \left[ q_1 + \left( \frac{1 - q_0 - q_1}{1 - C_{p,\theta} - C_{p,\theta} p (\cos \theta)^2} \right) \times \right. \\ & \left. \left( p \frac{C_{p,\theta}}{2} \left[ \frac{1}{(1 - p)^2} + \frac{\cos(2\theta)(1 + p^2) - 2p}{(1 - 2p \cos(2\theta) + p^2)^2} \right] - C_{p,\theta} p (\cos \theta)^2 \right) \right]^2. \end{aligned} \tag{18}$$

According to the likelihood function in Equation (16), the log-likelihood function $l = \ln L$ is given by

$$\begin{aligned} \ln L = & S_0 \ln q_0 + S_1 \ln q_1 + (n - S_0 - S_1) \ln (1 - q_0 - q_1) + S \ln p \\ & + 2 \sum_{Y_i \geq 2} \ln \cos(Y_i \theta) - (n - S_0 - S_1) \ln (C_{p,\theta}^{-1} - 1 - p (\cos \theta)^2). \end{aligned} \tag{19}$$

Hence, the MLEs of $q_0$ and $q_1$ are

$$\hat{q}_i = \frac{S_i}{n}, i = 0, 1, \tag{20}$$

In addition, the MLEs of $p$ and $\theta$ as well as $\hat{p}$ and $\hat{\theta}$ are the respective solutions to the following equations:

$$S(C_{p,\theta}^{-1} - 1 - p (\cos \theta)^2) - p(n - S_0 - S_1) \left( \frac{\partial C_{p,\theta}^{-1}}{\partial p} - (\cos \theta)^2 \right) = 0, \tag{21}$$

$$-2 \sum_{Y_i \geq 2} \tan(Y_i \theta) Y_i (C_{p,\theta}^{-1} - 1 - p (\cos \theta)^2) - (n - S_0 - S_1) \left( \frac{\partial C_{p,\theta}^{-1}}{\partial \theta} + 2p \cos \theta \sin \theta \right) = 0, \tag{22}$$

which can be solved numerically according to the Newton-type algorithm using the *nlm* function in [28]. Furthermore, parameters $\omega_1$ and $\omega_2$ in the ZOICG model in Equation (6) can be expressed by using $(q_0, q_1, p, \theta)$. Based on the one-to-one transformation in Equation (14), the MLEs of $\omega_1$ and $\omega_2$ are respectively given by

$$\omega_1 = q_0 - \frac{(1 - q_0 - q_1)C_{p,\theta}}{1 - C_{p,\theta} - C_{p,\theta}p(\cos\theta)^2}, \tag{23}$$

$$\omega_2 = q_1 - \frac{(1 - q_0 - q_1)C_{p,\theta}p(\cos\theta)^2}{1 - C_{p,\theta} - C_{p,\theta}p(\cos\theta)^2}. \tag{24}$$

### 2.3. The Fisher Information Matrix

The observed data log-likelihood function of the ZOICG model in Equation (15) can be expressed as

$$l_1(q_0, q_1, p, \theta, | \mathbf{Y}) = \sum_{i=1}^{n} \{ [\log q_0] I(y_i = 0) + [\log q_1] I(y_i = 1) + [\log(1 - q_0 - q_1) $$
$$+ y_i \log p + 2\log(\cos(y_i\theta)) - \log(C_{p,\theta}^{-1} - 1 - p(\cos\theta)^2)] I(y_i \geq 2) \}. \tag{25}$$

The score vector is denoted by

$$U(q_0, q_1, p, \theta) = \left( \frac{\partial l_1}{\partial q_0}, \frac{\partial l_1}{\partial q_1}, \frac{\partial l_1}{\partial p}, \frac{\partial l_1}{\partial \theta} \right)^{\top},$$

where

$$\frac{\partial l_1}{\partial q_0} = \sum_{i=1}^{n} \left[ \frac{1}{q_0} I(y_i = 0) - \frac{1}{(1 - q_0 - q_1)} I(y_i \geq 2) \right],$$

$$\frac{\partial l_1}{\partial q_1} = \sum_{i=1}^{n} \left[ \frac{1}{q_1} I(y_i = 1) - \frac{1}{(1 - q_0 - q_1)} I(y_i \geq 2) \right],$$

$$\frac{\partial l_1}{\partial p} = \sum_{i=1}^{n} \left[ \left( \frac{y_i}{p} - \frac{\frac{\partial}{\partial p}(C_{p,\theta}^{-1}) - (\cos\theta)^2}{C_{p,\theta}^{-1} - 1 - p(\cos\theta)^2} \right) I(y_i \geq 2) \right],$$

$$\frac{\partial l_1}{\partial \theta} = \sum_{i=1}^{n} \left[ \left( -\frac{2\sin(y_i\theta)y_i}{\cos(y_i\theta)} - \frac{\frac{\partial(C_{p,\theta}^{-1})}{\partial\theta} + 2p\cos(\theta)\sin(\theta)}{C_{p,\theta}^{-1} - 1 - p(\cos\theta)^2} \right) I(y_i \geq 2) \right].$$

The second derivatives are given by

$$\frac{\partial^2 l_1}{\partial q_0^2} = \sum_{i=1}^{n} \left[ -\frac{1}{q_0^2} I(y_i = 0) - \frac{1}{(1 - q_0 - q_1)^2} I(y_i \geq 2) \right],$$

$$\frac{\partial^2 l_1}{\partial q_1^2} = \sum_{i=1}^{n} \left[ -\frac{1}{q_1^2} I(y_i = 1) - \frac{1}{(1 - q_0 - q_1)^2} I(y_i \geq 2) \right],$$

$$\frac{\partial^2 l_1}{\partial p^2} = \sum_{i=1}^{n} \left[ \left( -\frac{y_i}{p^2} - \frac{\begin{array}{c}(C_{p,\theta}^{-1} - 1 - p(\cos\theta)^2)(\frac{\partial^2}{\partial p^2}(C_{p,\theta}^{-1})) \\ - (\frac{\partial}{\partial p}(C_{p,\theta}^{-1}) - (\cos\theta)^2)(\frac{\partial}{\partial p}(C_{p,\theta}^{-1}) - (\cos\theta)^2)\end{array}}{(C_{p,\theta}^{-1} - 1 - p(\cos\theta)^2)^2} \right) I(y_i \geq 2) \right],$$

$$\frac{\partial^2 l_1}{\partial \theta^2} =$$

$$\sum_{i=1}^{n} \left[ \left( -2\sec^2(y_i\theta)y_i^2 - \frac{\begin{array}{c}(C_{p,\theta}^{-1} - 1 - p(\cos\theta)^2)(\frac{\partial^2(C_{p,\theta}^{-1})}{\partial \theta^2} + \frac{\partial(2p\cos\theta\sin\theta)}{\partial \theta}) \\ -(\frac{\partial(C_{p,\theta}^{-1})}{\partial \theta} + 2p\cos\theta\sin\theta)(\frac{\partial(C_{p,\theta}^{-1})}{\partial \theta} + 2p\cos\theta\sin\theta)\end{array}}{(C_{p,\theta}^{-1} - 1 - p(\cos\theta)^2)^2} \right) I(y_i \geq 2) \right],$$

$$\frac{\partial^2 l_1}{\partial q_0 \partial q_1} = \sum_{i=1}^{n} \left[ -\frac{1}{(1 - q_0 - q_1)^2} I(y_i \geq 2) \right],$$

$$\frac{\partial^2 l_1}{\partial p \partial \theta} = \sum_{i=1}^{n} \left[ -\frac{\begin{array}{c}(C_{p,\theta}^{-1} - 1 - p(\cos\theta)^2)(\frac{\partial}{\partial \theta}(\frac{\partial(C_{p,\theta}^{-1} - (\cos\theta)^2)}{\partial p})) \\ -(\frac{\partial(C_{p,\theta}^{-1} - (\cos\theta)^2))}{\partial p})(\frac{\partial(C_{p,\theta}^{-1}) - (\cos\theta)^2)}{\partial \theta})\end{array}}{(C_{p,\theta}^{-1} - 1 - p(\cos\theta)^2)^2} I(y_i \geq 2) \right],$$

$$\frac{\partial^2 l_1}{\partial q_0 \partial p} = 0, \frac{\partial^2 l_1}{\partial q_0 \partial \theta} = 0, \frac{\partial^2 l_1}{\partial q_1 \partial p} = 0, \frac{\partial^2 l_1}{\partial q_1 \partial \theta} = 0.$$

Since

$$E[I(y_i = 0)] = q_0, E[I(y_i = 1)] = q_1, E[I(y_i \geq 2)] = 1 - q_0 - q_1.$$

Thus, the expected Fisher information matrix can be calculated as $J(q_0, q_1, p, \theta) = (J_{jj'})$, where

$$J_{11} = -E\left(\frac{\partial^2 l_1}{\partial q_0^2}\right) = \frac{n(1 - q_1)}{q_0(1 - q_0 - q_1)},$$

$$J_{22} = -E\left(\frac{\partial^2 l_1}{\partial q_1^2}\right) = \frac{n(1 - q_0)}{q_1(1 - q_0 - q_1)},$$

$$J_{33} = -E\left(\frac{\partial^2 l_1}{\partial p^2}\right) = n(1 - q_0 - q_1)k(p),$$

$$J_{44} = -E\left(\frac{\partial^2 l_1}{\partial \theta^2}\right) = n(1 - q_0 - q_1)k(\theta),$$

$$J_{12} = -E\left(\frac{\partial^2 l_1}{\partial q_0 \partial q_1}\right) = \frac{n}{(1 - q_0 - q_1)},$$

$$J_{34} = -E\left(\frac{\partial^2 l_1}{\partial p \partial \theta}\right) = n(1 - q_0 - q_1)k(p\theta),$$

$$J_{13} = -E\left(\frac{\partial^2 l_1}{\partial q_0 \partial p}\right) = 0, J_{14} = -E\left(\frac{\partial^2 l_1}{\partial q_0 \partial \theta}\right) = 0,$$

$$J_{23} = -E\left(\frac{\partial^2 l_1}{\partial q_1 \partial p}\right) = 0, J_{24} = -E\left(\frac{\partial^2 l_1}{\partial q_1 \partial \theta}\right) = 0.$$

Moreover, we have

$$k(p) = \frac{\frac{\partial(C_{p,\theta}^{-1})}{\partial p} - (\cos\theta)^2}{(C_{p,\theta}^{-1} - 1 - p(\cos\theta)^2)p} + \frac{\left[\begin{array}{c}(C_{p,\theta}^{-1} - 1 - p(\cos\theta)^2)(\frac{\partial^2}{\partial p^2}(C_{p,\theta}^{-1})) \\ - (\frac{\partial}{\partial p}(C_{p,\theta}^{-1}) - (\cos\theta)^2)^2\end{array}\right]}{(C_{p,\theta}^{-1} - 1 - p(\cos\theta)^2)^2},$$

$$k(p\theta) = \frac{\partial}{\partial\theta}\left(\frac{\frac{\partial(C_{p,\theta}^{-1})}{\partial p} - (\cos\theta)^2}{C_{p,\theta}^{-1} - 1 - p(\cos\theta)^2}\right),$$

$$k(\theta) = 2(C_{p,\theta}^{-1} - 1 - p(\cos\theta)^2)\left(\frac{p(1+p) - p(1-p)^3}{(1-p)^3}\right) +$$

$$\frac{\left[\begin{array}{c}(C_{p,\theta}^{-1} - 1 - p(\cos\theta)^2)(\frac{\partial^2(C_{p,\theta}^{-1})}{\partial\theta^2} + \frac{\partial(2p\cos\theta\sin\theta)}{\partial\theta}) \\ - (\frac{\partial(C_{p,\theta}^{-1})}{\partial\theta} + 2p\cos\theta\sin\theta)^2\end{array}\right]}{(C_{p,\theta}^{-1} - 1 - p(\cos\theta)^2)^2}.$$

Thus the expected Fisher information matrix of $(q_0, q_1, p, \theta)$ for one observation, when using a similar method to that in [11], is

$$H(q_0, q_1, p, \theta) = \begin{pmatrix} h_1 & 0 \\ 0 & h_2 \end{pmatrix}, \tag{26}$$

where

$$h_1 = \begin{pmatrix} \frac{1-q_1}{q_0(1-q_0-q_1)} & \frac{1}{(1-q_0-q_1)} \\ \frac{1}{(1-q_0-q_1)} & \frac{1-q_0}{q_1(1-q_0-q_1)} \end{pmatrix},$$

and

$$h_2 = \begin{pmatrix} (1-q_0-q_1)k(p) & (1-q_0-q_1)k(p\theta) \\ (1-q_0-q_1)k(p\theta) & (1-q_0-q_1)k(\theta) \end{pmatrix},$$

which implies that the two-group parameters $(q_0, q_1)$ and $(p, \theta)$ are independent, and this is useful information for applying the Bayesian inference in the next section.

## 3. Bayesian Inference

In this section, suitable priors for the ZOICG model in Equation (15) are derived by using the Bayesian framework.

### 3.1. Jeffreys Prior

Since Jeffreys prior [29,30] is proportional to the square root of the determinant of the Fisher information matrix, that for $(q_0, q_1, p, \theta)$ can be derived as follows:

$$\pi_J(q_0, q_1, p, \theta) = q_0^{-1/2} q_1^{-1/2}(1 - q_0 - q_1)^{1/2}(k(p)k(\theta) - k(p\theta)^2)^{1/2}, \tag{27}$$

where $0 \le q_0 \le 1$, $0 \le q_1 \le 1 - q_0$, $p \in (0, 1)$, and $\theta \in (0, \pi/2)$.

### 3.2. The Uniform Prior

The uniform prior on (0,1) is widely used as the prior distribution in Bayesian analysis, and that for $(q_0, q_1, p, \theta)$ is given by

$$\pi_U(q_0, q_1, p, \theta) = 1. \tag{28}$$

### 3.3. The Posterior Distribution

The joint posterior distributions of $(q_0, q_1, p, \theta)$ using the Jeffreys and uniform priors can be respectively expressed as

$$\pi_J(q_0, q_1, p, \theta \mid Y) = q_0^{S_0 - \frac{1}{2}} q_1^{S_1 - \frac{1}{2}} (1 - q_0 - q_1)^{n - S_0 - S_1 + \frac{1}{2}}$$
$$\frac{p^S \prod_{Y_i \geq 2} (\cos(Y_i \theta))^2 \left( k(p)k(\theta) - (k(p\theta))^2 \right)^{\frac{1}{2}}}{(C_{p,\theta}^{-1} - 1 - p(\cos \theta)^2)^{n - S_0 - S_1}}, \tag{29}$$

$$\pi_U(q_0, q_1, p, \theta \mid Y) = q_0^{S_0} q_1^{S_1} (1 - q_0 - q_1)^{n - S_0 - S_1}$$
$$\frac{p^S \prod_{Y_i \geq 2} (\cos(Y_i \theta))^2}{(C_{p,\theta}^{-1} - 1 - p(\cos \theta)^2)^{n - S_0 - S_1}}. \tag{30}$$

Bayesian inference is based on the posterior distribution of the parameter interest obtained analytically or via Markov chain Monte Carlo methods. In particular, the 95% Bayesian CI for the parameter interest is simply from the 2.5 percentile to the 97.5 percentile of the posterior distribution. For an asymmetric posterior (the usual case), the HPD interval is not the same as the Bayesian CI from the 2.5 percentile to the 97.5 percentile [31], and so in practice, their efficiencies are compared via their coverage probabilities (CPs) and average lengths (ALs). According to the joint posterior distribution of $(q_0, q_1, p, \theta)$ (Equations (29) and (30)), the marginal posterior distribution of $(q_0, q_1)$ using the Jeffreys and uniform priors are respectively given by

$$\pi_J(q_0, q_1 \mid Y) = q_0^{S_0 - \frac{1}{2}} q_1^{S_1 - \frac{1}{2}} (1 - q_0 - q_1)^{n - S_0 - S_1 + \frac{1}{2}}, \tag{31}$$

and

$$\pi_U(q_0, q_1 \mid Y) = q_0^{S_0} q_1^{S_1} (1 - q_0 - q_1)^{n - S_0 - S_1}. \tag{32}$$

These are Dirichlet distributions with shape parameters $(S_0 + \frac{1}{2}, S_1 + \frac{1}{2}, n - S_0 - S_1 + \frac{3}{2})$ and $(S_0 + 1, S_1 + 1, n - S_0 - S_1 + 1)$, respectively. The marginal posterior distribution of $p$ using the Jeffreys and uniform priors are respectively given by

$$\pi_J(p \mid \theta, Y) \propto \frac{p^S \left( k(p)k(\theta) - (k(p\theta))^2 \right)^{\frac{1}{2}}}{(C_{p,\theta}^{-1} - 1 - p(\cos \theta)^2)^{n - S_0 - S_1}}, \tag{33}$$

and

$$\pi_U(p \mid \theta, Y) \propto \frac{p^S}{(C_{p,\theta}^{-1} - 1 - p(\cos \theta)^2)^{n - S_0 - S_1}}. \tag{34}$$

Similarly, the marginal posterior distribution of $\theta$ using the Jeffreys and uniform priors are respectively given by

$$\pi_J(\theta \mid p, Y) \propto \frac{\prod_{Y_i \geq 2} (\cos(Y_i \theta))^2 \left( k(p)k(\theta) - (k(p\theta))^2 \right)^{\frac{1}{2}}}{(C_{p,\theta}^{-1} - 1 - p(\cos \theta)^2)^{n - S_0 - S_1}}, \tag{35}$$

and

$$\pi_U(\theta \mid p, Y) \propto \frac{\prod_{Y_i \geq 2} (\cos(Y_i \theta))^2}{(C_{p,\theta}^{-1} - 1 - p(\cos \theta)^2)^{n - S_0 - S_1}}. \tag{36}$$

Since it is evident that the closed forms of the marginal posterior distributions of $p$ and $\theta$ cannot be evaluated, various general-purpose simulation methods can be used to evaluate Equations (33)–(36) without requiring the complete closed-form expression. One popular method, the Metropolis–Hastings algorithm, was applied in the present

study. It was used to generate a large sequence of sampled values from the posterior distribution without fully specifying it, which then allowed us to estimate it [32].

The random walk Metropolis–Hastings algorithm is a special case of the Metropolis–Hastings algorithm that can be used to simulate the proposal $X_j^* = X_{j-1} + \epsilon_j, j = 0, 1, ..., N$, where $\epsilon_j$ is a random perturbation of $X_{j-1}$ that is simulated from a chosen probability distribution (e.g., a normal distribution with a mean of zero and a specified variance). Proposal $X_j^*$ is taken to be the next value in the chain $X_j$ with an acceptance probability that depends on the prior and data distributions (i.e., the likelihood function). The acceptance probability is the ratio of two posterior distributions, which can be calculated as follows:

$$R(X_{j-1}, X_j^*) = min\left( \frac{L(X_j^*)\pi(X_j^*)}{L(X_{j-1})\pi(X_{j-1})}, 1 \right). \tag{37}$$

If proposal $X_j^*$ is not taken, then $X_j$ is set to $X_{j-1}$ as the next value in the chain. This process is specified in Algorithm 1.

---

**Algorithm 1** The random walk Metropolis–Hastings algorithm.

1. Choose the trial position $X_j^* = X_{j-1} + \epsilon_j$, where $\epsilon_j$ is a random perturbation with a distribution $h$, which is symmetric (e.g., normal).
2. Calculate $R(X_{j-1}, X_j^*) = min\left( \frac{L(X_j^*)\pi(X_j^*)}{L(X_{j-1})\pi(X_{j-1})}, 1 \right)$.
3. Generate $U_j$ from $Uniform[0, 1]$.
4. If $U_j \leq R(X_{j-1}, X_j^*)$, then accept the change and let $X_j = X_j^*$; otherwise, set $X_j = X_{j-1}$.

---

### 3.4. The Bayesian CIs

In Bayesian analysis, interval estimates are based on the quantiles of the posterior distribution of a parameter. The interpretation of a credible interval is that it has a probability $1 - \alpha$ of containing the parameter because it is based on its posterior distribution. The most popular type of credible interval is an equal-tailed interval that is limited by the probabilities to the left of the lower limit ($\alpha/2$) and the right of the upper limit ($\alpha/2$). Thus, $(1 - \alpha)100\%$ can be obtained as follows:

$$\left(g_{(\alpha/2)}, g_{(1-\alpha/2)}\right), \tag{38}$$

where $g_{(\alpha/2)}$ and $g_{(1-\alpha/2)}$ are the $\alpha/2$ and $1 - \alpha/2$ quantiles of the function $g$, respectively.

Equal-tailed intervals are relatively easier to calculate than other ones, and they perform best when the posterior distribution is fairly symmetric. Since the marginal posterior distributions of $p$ and $\theta$ are dependent, and their closed forms do not exist, the Metropolis–Hastings steps within a Gibbs sampling algorithm can be used to estimate them [33]. Hence, it was applied to generate a large sequence of sampled values from the marginal posterior distribution in Equations (33)–(36). The procedure to generate samples and construct the Bayesian CIs for parameters $p$ and $\theta$ is detailed in Algorithm 2.

### 3.5. The HPD Intervals

When the posterior is not symmetric, it is best to use the HPD interval calculated by finding its lower and upper limits corresponding to the HPD region (i.e., the most probable region). The resulting interval is generally the most narrow possible interval that contains $(1 - \alpha)100\%$ of the posterior density. It can be obtained by using the Markov chain Monte Carlo method [34] and only requires a sample generated from the marginal posterior distributions of parameters $p$ and $\theta$. In the simulation and computation, the HPD intervals were computed by using the *HDInterval* package version 0.2.2 [35] in the R programming suite, as covered in Algorithm 3.

---

**Algorithm 2** Establishing the Bayesian CIs for parameters $p$ and $\theta$.

---

1. Take the initial values of $p$ and $\theta$ ($p_0$ and $\theta_0$, respectively).
2. Take the values $p_j$ and $\theta_j$ for $p$ and $\theta$ at the $j$th step for $j = 1, 2, ...N$, and then perform the following steps.
   (a) Generate $p_j^* = p_{j-1} + \epsilon_j$, where $\epsilon_j \sim N(0, \sigma_a^2)$.
   (b) Calculate $R(p_{j-1}, p_j^*) = min(r, 1)$, where $r = \frac{L(p_j^*, \theta_{j-1})\pi(p_j^*, \theta_{j-1})}{L(p_{j-1}, \theta_{j-1})\pi(p_{j-1}, \theta_{j-1})}$.
   (c) Generate $U_j \sim Uniform[0, 1]$.
   (d) If $U_j \leq R(p_{j-1}, p_j^*)$, then accept the change and let $p_j = p_j^*$; otherwise, set $p_j = p_{j-1}$.
   (e) Generate $\theta_j^* = \theta_{j-1} + \epsilon_j$, where $\epsilon_j \sim N(0, \sigma_b^2)$.
   (f) Calculate $R(\theta_{j-1}, \theta_j^*) = min(r, 1)$, where $r = \frac{L(p_j, \theta_j^*)\pi(p_j, \theta_j^*)}{L(p_j, \theta_{j-1})\pi(p_j, \theta_{j-1})}$.
   (g) Generate $U_j \sim Uniform[0, 1]$.
   (h) If $U_j \leq R(\theta_{j-1}, \theta_j^*)$, then accept the change and let $\theta_j = \theta_j^*$; otherwise, set $\theta_j = \theta_{j-1}$.
3. Repeat Step 2 $N$ times.
4. For posterior analysis, perform the following steps.
   (a) Calculate the Bayesian estimators of $g(p, \theta)$ by using $\frac{1}{N-M} \sum_{j=M+1}^{N} g(p_j, \theta_j)$, where $M$ is the number of burn-in samples.
   (b) Calculate the $100(1 - \alpha)\%$ CI as $(g_{(\alpha/2)}, g_{(1-\alpha/2)})$, where $g_{(\alpha/2)}$ and $g_{(1-\alpha/2)}$ are the $\frac{\alpha}{2}$-th and $1 - \frac{\alpha}{2}$-th quantiles of $g(p_j, \theta_j)$ for $j = M + 1, \ldots, N$, respectively.

---

**Algorithm 3** Establishing the HPD interval.

---

1. Take the initial values of $p$ and $\theta$ ($p_0$ and $\theta_0$, respectively).
2. Take the values $p_j$ and $\theta_j$ for $p$ and $\theta$, respectively, at the $j$th step for $j = 1, 2, ...N$, and then perform the following steps.
   (a) Generate $p_j^* = p_{j-1} + \epsilon_j$, where $\epsilon_j \sim N(0, \sigma_a^2)$.
   (b) Calculate $R(p_{j-1}, p_j^*) = min(r, 1)$, where $r = \frac{L(p_j^*, \theta_{j-1})\pi(p_j^*, \theta_{j-1})}{L(p_{j-1}, \theta_{j-1})\pi(p_{j-1}, \theta_{j-1})}$.
   (c) Generate $U_j \sim Uniform[0, 1]$.
   (d) If $U_j \leq R(p_{j-1}, p_j^*)$, then accept the change and let $p_j = p_j^*$; otherwise, set $p_j = p_{j-1}$.
   (e) Generate $\theta_j^* = \theta_{j-1} + \epsilon_j$, where $\epsilon_j \sim N(0, \sigma_b^2)$.
   (f) Calculate $R(\theta_{j-1}, \theta_j^*) = min(r, 1)$, where $r = \frac{L(p_j, \theta_j^*)\pi(p_j, \theta_j^*)}{L(p_j, \theta_{j-1})\pi(p_j, \theta_{j-1})}$.
   (g) Generate $U_j \sim Uniform[0, 1]$.
   (h) If $U_j \leq R(\theta_{j-1}, \theta_j^*)$, then accept the change and let $\theta_j = \theta_j^*$; otherwise, set $\theta_j = \theta_{j-1}$.
3. Repeat Step 2 $N$ times.
4. For posterior analysis, perform the following analysis.
   (a) Calculate the Bayesian estimators of $g(p, \theta)$ by using $\frac{1}{N-M} \sum_{j=M+1}^{N} g(p_j, \theta_j)$, where $M$ is the number of burn-in samples.
   (b) Calculate the HPD intervals $100(1 - \alpha)\%$ for each parameter by using the *HDInterval* package in the R programming suite [28].

---

## 4. Simulation Study

The performance of the ZOICG model in Equation (15) was measured by using Monte Carlo simulation. The sample size was set to $n = 30, 50$, or $100$ with the proviso that

$n - n_0 - n_1 > 0$. The probability of real zeros ($\omega_1$) was set to 0.1 or 0.2. The probability of real ones ($\omega_2$) was set to 0.1, 0.2, or 0.3 while $p, \theta$ was set to (0.5,0.7) or (0.9,1.5), and the nominal confidence level $\alpha$ was set to 0.95. All of the simulations were run 3000 times, and the samples were generated by using the Metropolis–Hastings algorithm with 10,000 samples and 3000 burn-ins. The criteria for comparing the efficiencies of the CIs were their CPs and ALs. For a particular scenario, the CI with a CP close to or greater than the nominal level of 0.95 and the shortest AL performed the best.

The CP and AL results are reported in Tables 1 and 2, respectively. For the interval estimation of the parameter $q_0$, the CPs of all five methods were close to the nominal level of 0.95, especially when the sample size was large. For the interval estimation of the parameter $q_1$, an equal-tailed, two-sided Bayesian CI based on the uniform prior provided CPs equal to or more than 0.95 in almost all cases, but when the sample size was large ($n = 100$), the CPs for all of the methods were similarly close to the nominal level of 0.95. For the interval estimation of parameter $p$, when $(p, \theta)$ was (0.5,0.7), only the equal-tailed two-sided Bayesian CI and HPD interval based on the uniform prior performed well (CP > 0.95), with the HPD interval based on the uniform prior having the shortest AL. However, when $(p, \theta)$ was (0.9,1.5), all five methods performed similarly. For the interval estimation of $\theta$, when $(p, \theta)$ was (0.5,0.7), all four Bayesian methods performed well (CP > 0.95), with the HPD interval based on the uniform prior providing the shortest AL. When $(p, \theta)$ was (0.9,1.5), none of the methods produced CPs equal to or more than 0.95, with the Wald CI providing the worst performance. In general, the ALs decreased when the sample size was increased and when $(p, \theta)$ was increased from (0.5,0.7) to (0.9,1.5).

**Table 1.** Coverage probability of parameter estimation for model in Equation (15).

| $n$ | $p, \theta$ | $\omega_1$ | $\omega_2$ | Method | $q_0$ | $q_1$ | $p$ | $\theta$ |
|---|---|---|---|---|---|---|---|---|
| 30 | 0.5,0.7 | 0.1 | 0.1 | Wald | 0.9490 | 0.9163 | 0.8475 | 0.3908 |
| | | | | B.uniform | 0.9347 | 0.9490 | 0.9800 | 0.9993 |
| | | | | HPD.uniform | 0.9353 | 0.9533 | 0.9633 | 0.9983 |
| | | | | B.Jeffreys | 0.9347 | 0.9577 | 0.7947 | 0.9983 |
| | | | | HPD.Jeffreys | 0.9347 | 0.9173 | 0.6957 | 0.9983 |
| | | | 0.2 | Wald | 0.9417 | 0.9030 | 0.8417 | 0.3846 |
| | | | | B.uniform | 0.9540 | 0.9497 | 0.9810 | 1.0000 |
| | | | | HPD.uniform | 0.9390 | 0.9493 | 0.9617 | 0.9993 |
| | | | | B.Jeffreys | 0.9313 | 0.9620 | 0.7613 | 0.9980 |
| | | | | HPD.Jeffreys | 0.9287 | 0.9157 | 0.6517 | 0.9977 |
| | | | 0.3 | Wald | 0.9293 | 0.9383 | 0.8498 | 0.4368 |
| | | | | B.uniform | 0.9493 | 0.9570 | 0.9837 | 0.9997 |
| | | | | HPD.uniform | 0.9420 | 0.9377 | 0.9680 | 0.9993 |
| | | | | B.Jeffreys | 0.9370 | 0.9360 | 0.7353 | 0.9980 |
| | | | | HPD.Jeffreys | 0.9157 | 0.9190 | 0.6263 | 0.9983 |
| | | 0.2 | 0.1 | Wald | 0.9293 | 0.9327 | 0.8442 | 0.4166 |
| | | | | B.uniform | 0.9390 | 0.9670 | 0.9787 | 0.9990 |
| | | | | HPD.uniform | 0.9383 | 0.9357 | 0.9593 | 0.9983 |
| | | | | B.Jeffreys | 0.9390 | 0.9360 | 0.7647 | 0.9987 |
| | | | | HPD.Jeffreys | 0.9387 | 0.9367 | 0.6657 | 0.9963 |
| | | | 0.2 | Wald | 0.9447 | 0.9330 | 0.8455 | 0.4271 |
| | | | | B.uniform | 0.9517 | 0.9637 | 0.9753 | 0.9997 |
| | | | | HPD.uniform | 0.9450 | 0.9327 | 0.9577 | 0.9997 |
| | | | | B.Jeffreys | 0.9303 | 0.9353 | 0.7267 | 0.9960 |
| | | | | HPD.Jeffreys | 0.9317 | 0.9353 | 0.6093 | 0.9963 |
| | | | 0.3 | Wald | 0.9403 | 0.9360 | 0.8421 | 0.4467 |
| | | | | B.uniform | 0.9540 | 0.9417 | 0.9843 | 1.0000 |
| | | | | HPD.uniform | 0.9507 | 0.9397 | 0.9653 | 0.9993 |
| | | | | B.Jeffreys | 0.9493 | 0.9473 | 0.7020 | 0.9983 |
| | | | | HPD.Jeffreys | 0.9260 | 0.9427 | 0.5737 | 0.9983 |

**Table 1.** *Cont.*

| $n$ | $p, \theta$ | $\omega_1$ | $\omega_2$ | Method | $q_0$ | $q_1$ | $p$ | $\theta$ |
|---|---|---|---|---|---|---|---|---|
| 30 | 0.9,1.5 | 0.1 | 0.1 | Wald | 0.9383 | 0.8193 | 0.9327 | 0.0137 |
| | | | | B.uniform | 0.9443 | 0.9637 | 0.9477 | 0.9167 |
| | | | | HPD.uniform | 0.9387 | 0.9240 | 0.9500 | 0.9033 |
| | | | | B.Jeffreys | 0.9383 | 0.9220 | 0.9460 | 0.9157 |
| | | | | HPD.Jeffreys | 0.9417 | 0.9440 | 0.9303 | 0.9110 |
| | | | 0.2 | Wald | 0.9317 | 0.9500 | 0.9377 | 0.0153 |
| | | | | B.uniform | 0.9477 | 0.9613 | 0.9427 | 0.9017 |
| | | | | HPD.uniform | 0.9323 | 0.9313 | 0.9483 | 0.8927 |
| | | | | B.Jeffreys | 0.9497 | 0.9340 | 0.9473 | 0.9023 |
| | | | | HPD.Jeffreys | 0.9307 | 0.9460 | 0.9363 | 0.8973 |
| | | | 0.3 | Wald | 0.8867 | 0.9050 | 0.9413 | 0.0183 |
| | | | | B.uniform | 0.9497 | 0.9580 | 0.9363 | 0.9010 |
| | | | | HPD.uniform | 0.9487 | 0.9577 | 0.9410 | 0.8990 |
| | | | | B.Jeffreys | 0.9503 | 0.9580 | 0.9480 | 0.8963 |
| | | | | HPD.Jeffreys | 0.9087 | 0.9127 | 0.9380 | 0.8923 |
| | | 0.2 | 0.1 | Wald | 0.9483 | 0.8070 | 0.9257 | 0.0207 |
| | | | | B.uniform | 0.9500 | 0.9753 | 0.9357 | 0.9017 |
| | | | | HPD.uniform | 0.9497 | 0.9323 | 0.9417 | 0.8953 |
| | | | | B.Jeffreys | 0.9497 | 0.9320 | 0.9430 | 0.9033 |
| | | | | HPD.Jeffreys | 0.9417 | 0.9453 | 0.9320 | 0.8927 |
| | | | 0.2 | Wald | 0.9157 | 0.9473 | 0.9337 | 0.0180 |
| | | | | B.uniform | 0.9540 | 0.9657 | 0.9340 | 0.8970 |
| | | | | HPD.uniform | 0.9270 | 0.9343 | 0.9403 | 0.8963 |
| | | | | B.Jeffreys | 0.9423 | 0.9357 | 0.9440 | 0.8983 |
| | | | | HPD.Jeffreys | 0.9283 | 0.9443 | 0.9350 | 0.8950 |
| | | | 0.3 | Wald | 0.9460 | 0.9120 | 0.9430 | 0.0200 |
| | | | | B.uniform | 0.9450 | 0.9567 | 0.9347 | 0.8690 |
| | | | | HPD.uniform | 0.9437 | 0.9563 | 0.9413 | 0.8763 |
| | | | | B.Jeffreys | 0.9460 | 0.9567 | 0.9463 | 0.8780 |
| | | | | HPD.Jeffreys | 0.9313 | 0.9210 | 0.9323 | 0.8803 |
| 50 | 0.5,0.7 | 0.1 | 0.1 | Wald | 0.9303 | 0.9243 | 0.8290 | 0.2558 |
| | | | | B.uniform | 0.9520 | 0.9473 | 0.9723 | 0.9953 |
| | | | | HPD.uniform | 0.9493 | 0.9487 | 0.9573 | 0.9910 |
| | | | | B.Jeffreys | 0.9480 | 0.9567 | 0.8747 | 0.9940 |
| | | | | HPD.Jeffreys | 0.9480 | 0.9303 | 0.8167 | 0.9930 |
| | | | 0.2 | Wald | 0.9457 | 0.9450 | 0.8382 | 0.3144 |
| | | | | B.uniform | 0.9567 | 0.9633 | 0.9790 | 0.9973 |
| | | | | HPD.uniform | 0.9487 | 0.9443 | 0.9673 | 0.9950 |
| | | | | B.Jeffreys | 0.9587 | 0.9457 | 0.8607 | 0.9973 |
| | | | | HPD.Jeffreys | 0.9510 | 0.9457 | 0.7883 | 0.9973 |
| | | | 0.3 | Wald | 0.9460 | 0.9627 | 0.8306 | 0.3353 |
| | | | | B.uniform | 0.9600 | 0.9620 | 0.9740 | 0.9967 |
| | | | | HPD.uniform | 0.9580 | 0.9507 | 0.9547 | 0.9957 |
| | | | | B.Jeffreys | 0.9583 | 0.9557 | 0.8270 | 0.9967 |
| | | | | HPD.Jeffreys | 0.9493 | 0.9373 | 0.7490 | 0.9977 |
| | | 0.2 | 0.1 | Wald | 0.9607 | 0.9303 | 0.8272 | 0.2866 |
| | | | | B.uniform | 0.9570 | 0.9623 | 0.9760 | 0.9973 |
| | | | | HPD.uniform | 0.9550 | 0.9607 | 0.9633 | 0.9933 |
| | | | | B.Jeffreys | 0.9487 | 0.9623 | 0.8610 | 0.9967 |
| | | | | HPD.Jeffreys | 0.9500 | 0.9353 | 0.7887 | 0.9957 |
| | | | 0.2 | Wald | 0.9490 | 0.9447 | 0.8527 | 0.3584 |
| | | | | B.uniform | 0.9550 | 0.9573 | 0.9797 | 0.9980 |
| | | | | HPD.uniform | 0.9497 | 0.9417 | 0.9640 | 0.9980 |
| | | | | B.Jeffreys | 0.9403 | 0.9460 | 0.8450 | 0.9973 |
| | | | | HPD.Jeffreys | 0.9427 | 0.9453 | 0.7657 | 0.9963 |
| | | | 0.3 | Wald | 0.9527 | 0.9530 | 0.8384 | 0.3670 |
| | | | | B.uniform | 0.9420 | 0.9500 | 0.9813 | 0.9983 |
| | | | | HPD.uniform | 0.9350 | 0.9350 | 0.9600 | 0.9983 |
| | | | | B.Jeffreys | 0.9407 | 0.9373 | 0.7993 | 0.9977 |
| | | | | HPD.Jeffreys | 0.9370 | 0.9203 | 0.7160 | 0.9967 |

**Table 1.** *Cont.*

| n | p, θ | ω₁ | ω₂ | Method | q₀ | q₁ | p | θ |
|---|---|---|---|---|---|---|---|---|
| 50 | 0.9,1.5 | 0.1 | 0.1 | Wald | 0.9403 | 0.8810 | 0.9307 | 0.0310 |
| | | | | B.uniform | 0.9550 | 0.9573 | 0.9460 | 0.9307 |
| | | | | HPD.uniform | 0.9363 | 0.9410 | 0.9463 | 0.9160 |
| | | | | B.Jeffreys | 0.9413 | 0.9397 | 0.9443 | 0.9357 |
| | | | | HPD.Jeffreys | 0.9403 | 0.9553 | 0.9363 | 0.9220 |
| | | | 0.2 | Wald | 0.9543 | 0.9353 | 0.9300 | 0.0317 |
| | | | | B.uniform | 0.9503 | 0.9477 | 0.9490 | 0.9273 |
| | | | | HPD.uniform | 0.9450 | 0.9357 | 0.9530 | 0.9120 |
| | | | | B.Jeffreys | 0.9527 | 0.9473 | 0.9490 | 0.9280 |
| | | | | HPD.Jeffreys | 0.9513 | 0.9353 | 0.9390 | 0.9220 |
| | | | 0.3 | Wald | 0.9263 | 0.9340 | 0.9270 | 0.0257 |
| | | | | B.uniform | 0.9453 | 0.9513 | 0.9390 | 0.9247 |
| | | | | HPD.uniform | 0.9460 | 0.9367 | 0.9430 | 0.9177 |
| | | | | B.Jeffreys | 0.9510 | 0.9443 | 0.9443 | 0.9243 |
| | | | | HPD.Jeffreys | 0.9350 | 0.9440 | 0.9330 | 0.9127 |
| | | 0.2 | 0.1 | Wald | 0.9517 | 0.8690 | 0.9190 | 0.0263 |
| | | | | B.uniform | 0.9533 | 0.9677 | 0.9373 | 0.9237 |
| | | | | HPD.uniform | 0.9523 | 0.9403 | 0.9380 | 0.9137 |
| | | | | B.Jeffreys | 0.9553 | 0.9387 | 0.9400 | 0.9263 |
| | | | | HPD.Jeffreys | 0.9340 | 0.9497 | 0.9317 | 0.9097 |
| | | | 0.2 | Wald | 0.9173 | 0.9270 | 0.9313 | 0.0284 |
| | | | | B.uniform | 0.9487 | 0.9407 | 0.9413 | 0.9287 |
| | | | | HPD.uniform | 0.9407 | 0.9263 | 0.9427 | 0.9170 |
| | | | | B.Jeffreys | 0.9470 | 0.9360 | 0.9423 | 0.9280 |
| | | | | HPD.Jeffreys | 0.9213 | 0.9267 | 0.9337 | 0.9150 |
| | | | 0.3 | Wald | 0.9303 | 0.9417 | 0.9337 | 0.0280 |
| | | | | B.uniform | 0.9553 | 0.9603 | 0.9430 | 0.9050 |
| | | | | HPD.uniform | 0.9390 | 0.9453 | 0.9427 | 0.9010 |
| | | | | B.Jeffreys | 0.9427 | 0.9513 | 0.9447 | 0.9063 |
| | | | | HPD.Jeffreys | 0.9337 | 0.9520 | 0.9310 | 0.8987 |
| 100 | 0.5,0.7 | 0.1 | 0.1 | Wald | 0.9377 | 0.9527 | 0.7251 | 0.0807 |
| | | | | B.uniform | 0.9517 | 0.9587 | 0.9763 | 0.9777 |
| | | | | HPD.uniform | 0.9430 | 0.9543 | 0.9630 | 0.9733 |
| | | | | B.Jeffreys | 0.9510 | 0.9527 | 0.9417 | 0.9837 |
| | | | | HPD.Jeffreys | 0.9463 | 0.9503 | 0.9143 | 0.9823 |
| | | | 0.2 | Wald | 0.9547 | 0.9607 | 0.7493 | 0.1100 |
| | | | | B.uniform | 0.9557 | 0.9593 | 0.9727 | 0.9837 |
| | | | | HPD.uniform | 0.9523 | 0.9557 | 0.9597 | 0.9787 |
| | | | | B.Jeffreys | 0.9520 | 0.9603 | 0.9290 | 0.9873 |
| | | | | HPD.Jeffreys | 0.9523 | 0.9493 | 0.8970 | 0.9857 |
| | | | 0.3 | Wald | 0.9507 | 0.9483 | 0.7827 | 0.1440 |
| | | | | B.uniform | 0.9567 | 0.9533 | 0.9787 | 0.9867 |
| | | | | HPD.uniform | 0.9503 | 0.9503 | 0.9653 | 0.9833 |
| | | | | B.Jeffreys | 0.9470 | 0.9473 | 0.9230 | 0.9883 |
| | | | | HPD.Jeffreys | 0.9463 | 0.9473 | 0.8860 | 0.9867 |
| | | 0.2 | 0.1 | Wald | 0.9473 | 0.9333 | 0.7569 | 0.1027 |
| | | | | B.uniform | 0.9523 | 0.9530 | 0.9710 | 0.9780 |
| | | | | HPD.uniform | 0.9467 | 0.9537 | 0.9587 | 0.9767 |
| | | | | B.Jeffreys | 0.9513 | 0.9567 | 0.9363 | 0.9820 |
| | | | | HPD.Jeffreys | 0.9450 | 0.9443 | 0.9040 | 0.9813 |
| | | | 0.2 | Wald | 0.9483 | 0.9567 | 0.7883 | 0.1517 |
| | | | | B.uniform | 0.9493 | 0.9463 | 0.9787 | 0.9827 |
| | | | | HPD.uniform | 0.9480 | 0.9460 | 0.9680 | 0.9803 |
| | | | | B.Jeffreys | 0.9493 | 0.9540 | 0.9277 | 0.9853 |
| | | | | HPD.Jeffreys | 0.9477 | 0.9420 | 0.8897 | 0.9837 |
| | | | 0.3 | Wald | 0.9420 | 0.9440 | 0.8145 | 0.1952 |
| | | | | B.uniform | 0.9490 | 0.9497 | 0.9787 | 0.9897 |
| | | | | HPD.uniform | 0.9410 | 0.9447 | 0.9663 | 0.9873 |
| | | | | B.Jeffreys | 0.9373 | 0.9443 | 0.9163 | 0.9917 |
| | | | | HPD.Jeffreys | 0.9377 | 0.9437 | 0.8760 | 0.9897 |

**Table 1.** *Cont.*

| n | p, θ | ω₁ | ω₂ | Method | q₀ | q₁ | p | θ |
|---|---|---|---|---|---|---|---|---|
| 100 | 0.9,1.5 | 0.1 | 0.1 | Wald | 0.9370 | 0.9343 | 0.9080 | 0.0627 |
| | | | | B.uniform | 0.9473 | 0.9543 | 0.9283 | 0.9080 |
| | | | | HPD.uniform | 0.9337 | 0.9587 | 0.9270 | 0.8853 |
| | | | | B.Jeffreys | 0.9370 | 0.9593 | 0.9287 | 0.9140 |
| | | | | HPD.Jeffreys | 0.9353 | 0.9297 | 0.9217 | 0.8963 |
| | | | 0.2 | Wald | 0.9313 | 0.9293 | 0.9043 | 0.0597 |
| | | | | B.uniform | 0.9403 | 0.9487 | 0.9300 | 0.9100 |
| | | | | HPD.uniform | 0.9437 | 0.9480 | 0.9287 | 0.8957 |
| | | | | B.Jeffreys | 0.9493 | 0.9517 | 0.9367 | 0.9153 |
| | | | | HPD.Jeffreys | 0.9367 | 0.9327 | 0.9290 | 0.8980 |
| | | | 0.3 | Wald | 0.9417 | 0.9527 | 0.9093 | 0.0490 |
| | | | | B.uniform | 0.9487 | 0.9530 | 0.9430 | 0.9223 |
| | | | | HPD.uniform | 0.9407 | 0.9527 | 0.9410 | 0.9050 |
| | | | | B.Jeffreys | 0.9490 | 0.9527 | 0.9407 | 0.9267 |
| | | | | HPD.Jeffreys | 0.9400 | 0.9430 | 0.9343 | 0.9083 |
| | | 0.2 | 0.1 | Wald | 0.9573 | 0.9477 | 0.9077 | 0.0564 |
| | | | | B.uniform | 0.9500 | 0.9563 | 0.9200 | 0.9097 |
| | | | | HPD.uniform | 0.9510 | 0.9630 | 0.9200 | 0.8920 |
| | | | | B.Jeffreys | 0.9573 | 0.9633 | 0.9267 | 0.9220 |
| | | | | HPD.Jeffreys | 0.9487 | 0.9393 | 0.9217 | 0.9083 |
| | | | 0.2 | Wald | 0.9370 | 0.9343 | 0.9077 | 0.0427 |
| | | | | B.uniform | 0.9440 | 0.9583 | 0.9390 | 0.9203 |
| | | | | HPD.uniform | 0.9423 | 0.9523 | 0.9353 | 0.9023 |
| | | | | B.Jeffreys | 0.9487 | 0.9580 | 0.9393 | 0.9213 |
| | | | | HPD.Jeffreys | 0.9367 | 0.9377 | 0.9340 | 0.9057 |
| | | | 0.3 | Wald | 0.9503 | 0.9517 | 0.9193 | 0.0424 |
| | | | | B.uniform | 0.9557 | 0.9523 | 0.9477 | 0.9380 |
| | | | | HPD.uniform | 0.9453 | 0.9487 | 0.9463 | 0.9183 |
| | | | | B.Jeffreys | 0.9490 | 0.9520 | 0.9430 | 0.9417 |
| | | | | HPD.Jeffreys | 0.9470 | 0.9420 | 0.9357 | 0.9237 |

**Table 2.** Average length of parameter estimation for model in Equation (15).

| n | p, θ | ω₁ | ω₂ | Method | q₀ | q₁ | p | θ |
|---|---|---|---|---|---|---|---|---|
| 30 | 0.5,0.7 | 0.1 | 0.1 | Wald | 0.3349 | 0.3084 | 0.8341 | 0.2746 |
| | | | | B.uniform | 0.3185 | 0.2904 | 0.6684 | 1.2500 |
| | | | | HPD.uniform | 0.3165 | 0.2867 | 0.6595 | 1.1740 |
| | | | | B.Jeffreys | 0.3216 | 0.2880 | 0.6609 | 1.2798 |
| | | | | HPD.Jeffreys | 0.3197 | 0.2839 | 0.5983 | 1.2230 |
| | | | 0.2 | Wald | 0.3468 | 0.3324 | 0.8547 | 0.2753 |
| | | | | B.uniform | 0.3262 | 0.3103 | 0.6792 | 1.2850 |
| | | | | HPD.uniform | 0.3247 | 0.3079 | 0.6698 | 1.2162 |
| | | | | B.Jeffreys | 0.3292 | 0.3098 | 0.6479 | 1.3118 |
| | | | | HPD.Jeffreys | 0.3277 | 0.3071 | 0.5759 | 1.2614 |
| | | | 0.3 | Wald | 0.3518 | 0.3472 | 0.9043 | 0.3797 |
| | | | | B.uniform | 0.3289 | 0.3235 | 0.6978 | 1.3176 |
| | | | | HPD.uniform | 0.3276 | 0.3219 | 0.6878 | 1.2556 |
| | | | | B.Jeffreys | 0.3310 | 0.3245 | 0.6431 | 1.3438 |
| | | | | HPD.Jeffreys | 0.3297 | 0.3228 | 0.5637 | 1.2985 |
| | | 0.2 | 0.1 | Wald | 0.3271 | 0.2985 | 0.8682 | 0.2811 |
| | | | | B.uniform | 0.3130 | 0.2824 | 0.6793 | 1.2818 |
| | | | | HPD.uniform | 0.3108 | 0.2782 | 0.6697 | 1.2125 |
| | | | | B.Jeffreys | 0.3164 | 0.2792 | 0.6510 | 1.3097 |
| | | | | HPD.Jeffreys | 0.3142 | 0.2745 | 0.5804 | 1.2561 |
| | | | 0.2 | Wald | 0.3427 | 0.3262 | 0.8995 | 0.2862 |
| | | | | B.uniform | 0.3236 | 0.3052 | 0.6908 | 1.3139 |
| | | | | HPD.uniform | 0.3220 | 0.3025 | 0.6806 | 1.2497 |
| | | | | B.Jeffreys | 0.3266 | 0.3041 | 0.6364 | 1.3390 |
| | | | | HPD.Jeffreys | 0.3249 | 0.3012 | 0.5561 | 1.2929 |
| | | | 0.3 | Wald | 0.3510 | 0.3445 | 0.9318 | 0.3425 |
| | | | | B.uniform | 0.3289 | 0.3210 | 0.7071 | 1.3428 |
| | | | | HPD.uniform | 0.3275 | 0.3192 | 0.6964 | 1.2888 |
| | | | | B.Jeffreys | 0.3312 | 0.3217 | 0.6271 | 1.3666 |
| | | | | HPD.Jeffreys | 0.3299 | 0.3198 | 0.5372 | 1.3280 |

**Table 2.** *Cont.*

| *n* | *p, θ* | *ω₁* | *ω₂* | Method | *q₀* | *q₁* | *p* | *θ* |
|---|---|---|---|---|---|---|---|---|
| 30 | 0.9,1.5 | 0.1 | 0.1 | Wald | 0.3019 | 0.2035 | 0.0946 | 8.4846 |
| | | | | B.uniform | 0.2857 | 0.2093 | 0.0921 | 0.5622 |
| | | | | HPD.uniform | 0.2817 | 0.1992 | 0.0905 | 0.4801 |
| | | | | B.Jeffreys | 0.2828 | 0.1979 | 0.0894 | 0.5176 |
| | | | | HPD.Jeffreys | 0.2784 | 0.1867 | 0.0877 | 0.4461 |
| | | | 0.2 | Wald | 0.2953 | 0.2784 | 0.1027 | 69.7431 |
| | | | | B.uniform | 0.2796 | 0.2659 | 0.1010 | 0.6884 |
| | | | | HPD.uniform | 0.2751 | 0.2605 | 0.0991 | 0.5893 |
| | | | | B.Jeffreys | 0.2761 | 0.2610 | 0.0975 | 0.6627 |
| | | | | HPD.Jeffreys | 0.2711 | 0.2550 | 0.0955 | 0.5653 |
| | | | 0.3 | Wald | 0.2855 | 0.3208 | 0.1118 | 235.6500 |
| | | | | B.uniform | 0.2714 | 0.3006 | 0.1105 | 0.8728 |
| | | | | HPD.uniform | 0.2664 | 0.2976 | 0.1083 | 0.7526 |
| | | | | B.Jeffreys | 0.2672 | 0.2991 | 0.1062 | 0.8232 |
| | | | | HPD.Jeffreys | 0.2616 | 0.2957 | 0.1039 | 0.7242 |
| | | 0.2 | 0.1 | Wald | 0.3307 | 0.2008 | 0.1023 | 73.9848 |
| | | | | B.uniform | 0.3098 | 0.2076 | 0.1004 | 0.7126 |
| | | | | HPD.uniform | 0.3074 | 0.1974 | 0.0986 | 0.5966 |
| | | | | B.Jeffreys | 0.3091 | 0.1960 | 0.0970 | 0.6675 |
| | | | | HPD.Jeffreys | 0.3065 | 0.1845 | 0.0951 | 0.5749 |
| | | | 0.2 | Wald | 0.3254 | 0.2784 | 0.1125 | 161.9909 |
| | | | | B.uniform | 0.3044 | 0.2657 | 0.1111 | 0.8977 |
| | | | | HPD.uniform | 0.3016 | 0.2604 | 0.1090 | 0.7746 |
| | | | | B.Jeffreys | 0.3033 | 0.2608 | 0.1067 | 0.8484 |
| | | | | HPD.Jeffreys | 0.3002 | 0.2547 | 0.1044 | 0.7316 |
| | | | 0.3 | Wald | 0.3196 | 0.3218 | 0.1255 | 312.1138 |
| | | | | B.uniform | 0.2995 | 0.3013 | 0.1251 | 1.0833 |
| | | | | HPD.uniform | 0.2964 | 0.2983 | 0.1225 | 0.9495 |
| | | | | B.Jeffreys | 0.2979 | 0.3000 | 0.1193 | 1.0410 |
| | | | | HPD.Jeffreys | 0.2946 | 0.2967 | 0.1163 | 0.9198 |
| 50 | 0.5,0.7 | 0.1 | 0.1 | Wald | 0.2594 | 0.2406 | 0.6838 | 0.2925 |
| | | | | B.uniform | 0.2517 | 0.2317 | 0.6060 | 1.0605 |
| | | | | HPD.uniform | 0.2504 | 0.2297 | 0.5990 | 0.9636 |
| | | | | B.Jeffreys | 0.2534 | 0.2306 | 0.6564 | 1.0967 |
| | | | | HPD.Jeffreys | 0.2521 | 0.2285 | 0.6228 | 1.0169 |
| | | | 0.2 | Wald | 0.2700 | 0.2605 | 0.7412 | 115.0792 |
| | | | | B.uniform | 0.2600 | 0.2495 | 0.6312 | 1.1346 |
| | | | | HPD.uniform | 0.2590 | 0.2481 | 0.6236 | 1.0460 |
| | | | | B.Jeffreys | 0.2615 | 0.2495 | 0.6682 | 1.1765 |
| | | | | HPD.Jeffreys | 0.2605 | 0.2480 | 0.6244 | 1.1024 |
| | | | 0.3 | Wald | 0.2743 | 0.2710 | 0.7732 | 0.2561 |
| | | | | B.uniform | 0.2631 | 0.2593 | 0.6445 | 1.1893 |
| | | | | HPD.uniform | 0.2621 | 0.2582 | 0.6364 | 1.0999 |
| | | | | B.Jeffreys | 0.2643 | 0.2599 | 0.6646 | 1.2242 |
| | | | | HPD.Jeffreys | 0.2634 | 0.2588 | 0.6134 | 1.1535 |
| | | 0.2 | 0.1 | Wald | 0.2536 | 0.2344 | 0.7172 | 0.3026 |
| | | | | B.uniform | 0.2472 | 0.2262 | 0.6242 | 1.1240 |
| | | | | HPD.uniform | 0.2457 | 0.2240 | 0.6169 | 1.0287 |
| | | | | B.Jeffreys | 0.2489 | 0.2248 | 0.6630 | 1.1628 |
| | | | | HPD.Jeffreys | 0.2473 | 0.2225 | 0.6212 | 1.0832 |
| | | | 0.2 | Wald | 0.2660 | 0.2554 | 0.7880 | 0.2916 |
| | | | | B.uniform | 0.2568 | 0.2447 | 0.6472 | 1.1873 |
| | | | | HPD.uniform | 0.2556 | 0.2431 | 0.6394 | 1.0995 |
| | | | | B.Jeffreys | 0.2584 | 0.2445 | 0.6718 | 1.2230 |
| | | | | HPD.Jeffreys | 0.2573 | 0.2429 | 0.6205 | 1.1520 |
| | | | 0.3 | Wald | 0.2731 | 0.2686 | 0.8157 | 0.2645 |
| | | | | B.uniform | 0.2623 | 0.2568 | 0.6640 | 1.2355 |
| | | | | HPD.uniform | 0.2613 | 0.2557 | 0.6549 | 1.1573 |
| | | | | B.Jeffreys | 0.2635 | 0.2573 | 0.6669 | 1.2646 |
| | | | | HPD.Jeffreys | 0.2625 | 0.2561 | 0.6076 | 1.2048 |

**Table 2.** *Cont.*

| n | p, θ | ω₁ | ω₂ | Method | q₀ | q₁ | p | θ |
|---|---|---|---|---|---|---|---|---|
| 50 | 0.9,1.5 | 0.1 | 0.1 | Wald | 0.2376 | 0.1622 | 0.0714 | 470.0039 |
| | | | | B.uniform | 0.2289 | 0.1639 | 0.0684 | 0.1848 |
| | | | | HPD.uniform | 0.2268 | 0.1586 | 0.0673 | 0.1589 |
| | | | | B.Jeffreys | 0.2276 | 0.1584 | 0.0672 | 0.1689 |
| | | | | HPD.Jeffreys | 0.2254 | 0.1527 | 0.0661 | 0.1522 |
| | | | 0.2 | Wald | 0.2317 | 0.2178 | 0.0768 | 327.0361 |
| | | | | B.uniform | 0.2235 | 0.2113 | 0.0740 | 0.2479 |
| | | | | HPD.uniform | 0.2212 | 0.2084 | 0.0728 | 0.2125 |
| | | | | B.Jeffreys | 0.2221 | 0.2090 | 0.0724 | 0.2131 |
| | | | | HPD.Jeffreys | 0.2196 | 0.2060 | 0.0712 | 0.1884 |
| | | | 0.3 | Wald | 0.2235 | 0.2514 | 0.0842 | 320.3098 |
| | | | | B.uniform | 0.2163 | 0.2412 | 0.0810 | 0.3379 |
| | | | | HPD.uniform | 0.2137 | 0.2395 | 0.0796 | 0.2865 |
| | | | | B.Jeffreys | 0.2143 | 0.2406 | 0.0788 | 0.3010 |
| | | | | HPD.Jeffreys | 0.2115 | 0.2388 | 0.0774 | 0.2598 |
| | | 0.2 | 0.1 | Wald | 0.2581 | 0.1604 | 0.0769 | 9.1145 |
| | | | | B.uniform | 0.2474 | 0.1625 | 0.0740 | 0.2402 |
| | | | | HPD.uniform | 0.2459 | 0.1571 | 0.0728 | 0.2140 |
| | | | | B.Jeffreys | 0.2472 | 0.1567 | 0.0723 | 0.2032 |
| | | | | HPD.Jeffreys | 0.2456 | 0.1509 | 0.0711 | 0.1787 |
| | | | 0.2 | Wald | 0.2548 | 0.2180 | 0.0844 | 104.7380 |
| | | | | B.uniform | 0.2443 | 0.2116 | 0.0814 | 0.3295 |
| | | | | HPD.uniform | 0.2427 | 0.2087 | 0.0801 | 0.2791 |
| | | | | B.Jeffreys | 0.2439 | 0.2093 | 0.0792 | 0.3101 |
| | | | | HPD.Jeffreys | 0.2422 | 0.2063 | 0.0779 | 0.2660 |
| | | | 0.3 | Wald | 0.2498 | 0.2511 | 0.0940 | 967.9396 |
| | | | | B.uniform | 0.2397 | 0.2409 | 0.0909 | 0.5202 |
| | | | | HPD.uniform | 0.2379 | 0.2392 | 0.0894 | 0.4362 |
| | | | | B.Jeffreys | 0.2391 | 0.2403 | 0.0883 | 0.4688 |
| | | | | HPD.Jeffreys | 0.2372 | 0.2385 | 0.0866 | 0.4011 |
| 100 | 0.5,0.7 | 0.1 | 0.1 | Wald | 0.1843 | 0.1715 | 0.4658 | 0.1680 |
| | | | | B.uniform | 0.1814 | 0.1681 | 0.4927 | 0.6192 |
| | | | | HPD.uniform | 0.1807 | 0.1670 | 0.4879 | 0.5403 |
| | | | | B.Jeffreys | 0.1821 | 0.1677 | 0.5554 | 0.6432 |
| | | | | HPD.Jeffreys | 0.1813 | 0.1667 | 0.5451 | 0.5618 |
| | | | 0.2 | Wald | 0.1916 | 0.1851 | 0.5029 | 0.1803 |
| | | | | B.uniform | 0.1880 | 0.1809 | 0.5165 | 0.7212 |
| | | | | HPD.uniform | 0.1873 | 0.1801 | 0.5114 | 0.6294 |
| | | | | B.Jeffreys | 0.1886 | 0.1810 | 0.5843 | 0.7510 |
| | | | | HPD.Jeffreys | 0.1879 | 0.1802 | 0.5719 | 0.6608 |
| | | | 0.3 | Wald | 0.1949 | 0.1927 | 0.5561 | 0.1967 |
| | | | | B.uniform | 0.1907 | 0.1883 | 0.5420 | 0.8150 |
| | | | | HPD.uniform | 0.1901 | 0.1876 | 0.5366 | 0.7153 |
| | | | | B.Jeffreys | 0.1913 | 0.1887 | 0.6119 | 0.8606 |
| | | | | HPD.Jeffreys | 0.1906 | 0.1880 | 0.5959 | 0.7668 |
| | | 0.2 | 0.1 | Wald | 0.1801 | 0.1672 | 0.5028 | 0.1798 |
| | | | | B.uniform | 0.1777 | 0.1641 | 0.5150 | 0.7007 |
| | | | | HPD.uniform | 0.1769 | 0.1630 | 0.5100 | 0.6115 |
| | | | | B.Jeffreys | 0.1783 | 0.1635 | 0.5840 | 0.7322 |
| | | | | HPD.Jeffreys | 0.1775 | 0.1624 | 0.5719 | 0.6394 |
| | | | 0.2 | Wald | 0.1892 | 0.1824 | 0.5620 | 0.1971 |
| | | | | B.uniform | 0.1858 | 0.1784 | 0.5480 | 0.8221 |
| | | | | HPD.uniform | 0.1851 | 0.1776 | 0.5424 | 0.7248 |
| | | | | B.Jeffreys | 0.1864 | 0.1784 | 0.6165 | 0.8547 |
| | | | | HPD.Jeffreys | 0.1857 | 0.1775 | 0.5988 | 0.7657 |
| | | | 0.3 | Wald | 0.1940 | 0.1913 | 0.6138 | 0.2557 |
| | | | | B.uniform | 0.1900 | 0.1869 | 0.5742 | 0.9390 |
| | | | | HPD.uniform | 0.1894 | 0.1862 | 0.5681 | 0.8384 |
| | | | | B.Jeffreys | 0.1905 | 0.1872 | 0.6395 | 0.9788 |
| | | | | HPD.Jeffreys | 0.1899 | 0.1865 | 0.6161 | 0.8862 |

**Table 2.** *Cont.*

| n | p, θ | ω₁ | ω₂ | Method | q₀ | q₁ | p | θ |
|---|---|---|---|---|---|---|---|---|
| 100 | 0.9,1.5 | 0.1 | 0.1 | Wald | 0.1688 | 0.1166 | 0.0497 | 912.8982 |
| | | | | B.uniform | 0.1655 | 0.1170 | 0.0498 | 0.1298 |
| | | | | HPD.uniform | 0.1644 | 0.1148 | 0.0490 | 0.1144 |
| | | | | B.Jeffreys | 0.1650 | 0.1150 | 0.0500 | 0.1266 |
| | | | | HPD.Jeffreys | 0.1640 | 0.1128 | 0.0489 | 0.1135 |
| | | | 0.2 | Wald | 0.1644 | 0.1558 | 0.0534 | 699.7486 |
| | | | | B.uniform | 0.1613 | 0.1532 | 0.0527 | 0.1400 |
| | | | | HPD.uniform | 0.1602 | 0.1519 | 0.0515 | 0.1246 |
| | | | | B.Jeffreys | 0.1609 | 0.1524 | 0.0512 | 0.1267 |
| | | | | HPD.Jeffreys | 0.1597 | 0.1512 | 0.0504 | 0.1135 |
| | | | 0.3 | Wald | 0.1593 | 0.1788 | 0.0576 | 592.8977 |
| | | | | B.uniform | 0.1565 | 0.1749 | 0.0556 | 0.1255 |
| | | | | HPD.uniform | 0.1553 | 0.1740 | 0.0547 | 0.1104 |
| | | | | B.Jeffreys | 0.1558 | 0.1747 | 0.0543 | 0.1186 |
| | | | | HPD.Jeffreys | 0.1546 | 0.1738 | 0.0534 | 0.1045 |
| | | 0.2 | 0.1 | Wald | 0.1837 | 0.1163 | 0.0535 | 243.9278 |
| | | | | B.uniform | 0.1796 | 0.1167 | 0.0514 | 0.1326 |
| | | | | HPD.uniform | 0.1788 | 0.1145 | 0.0499 | 0.1149 |
| | | | | B.Jeffreys | 0.1796 | 0.1146 | 0.0512 | 0.1216 |
| | | | | HPD.Jeffreys | 0.1788 | 0.1124 | 0.0500 | 0.1145 |
| | | | 0.2 | Wald | 0.1808 | 0.1559 | 0.0579 | 87.2649 |
| | | | | B.uniform | 0.1769 | 0.1534 | 0.0552 | 0.1219 |
| | | | | HPD.uniform | 0.1760 | 0.1522 | 0.0543 | 0.1094 |
| | | | | B.Jeffreys | 0.1767 | 0.1526 | 0.0549 | 0.1207 |
| | | | | HPD.Jeffreys | 0.1759 | 0.1512 | 0.0540 | 0.1088 |
| | | | 0.3 | Wald | 0.1777 | 0.1786 | 0.0637 | 95.0276 |
| | | | | B.uniform | 0.1739 | 0.1748 | 0.0606 | 0.1391 |
| | | | | HPD.uniform | 0.1730 | 0.1739 | 0.0597 | 0.1253 |
| | | | | B.Jeffreys | 0.1737 | 0.1746 | 0.0597 | 0.1326 |
| | | | | HPD.Jeffreys | 0.1728 | 0.1737 | 0.0587 | 0.1226 |

## 5. The Efficacies of the Methods with Real Data

The number of new COVID-19 cases and deaths reported each day by country is available from the European Centre for Disease Prevention and Control (ECDC; accessed on 13 September 2022 https://www.ecdc.europa.eu/en/publications-data/data-daily-new-cases-covid-19-eueea-country). In this empirical study, the number of new COVID-19 deaths per day in Luxembourg from 24 February 2020 to 31 December 2020, which contained 312 days, 167 days with 0 deaths, and 47 days with 1 death, are presented in Table 3 and Figure 1. From the descriptive statistics for the dataset (Table 4), the mean and variance were 1.6314 and 5.9570, respectively, which were used to calculate the index of dispersion (the variance divided by the mean) to be 3.6514. Since the index of dispersion was larger than one, this dataset was clearly overdispersed. The appropriate model was checked by comparing the AIC and corrected AIC (AICc) values of nine distributions: ZOICG, ZICG, CG, ZIP, ZIG, ZINB, Poisson, geometric, and NB (Table 5). Those of the ZOICG were very similar (1038.372 and 1038.502) and the lowest, thereby inferring that it provided the best fit for the data.

**Table 3.** The number of COVID-19 daily new deaths in Luxembourg in 2020.

| The Number of Daily New Deaths | 0 | 1 | 2 | 3 | 4 | 5 | 6 | 7 | 8 | 9 | 10 | 11 |
|---|---|---|---|---|---|---|---|---|---|---|---|---|
| Count | 167 | 47 | 17 | 20 | 15 | 9 | 13 | 15 | 4 | 3 | 1 | 1 |

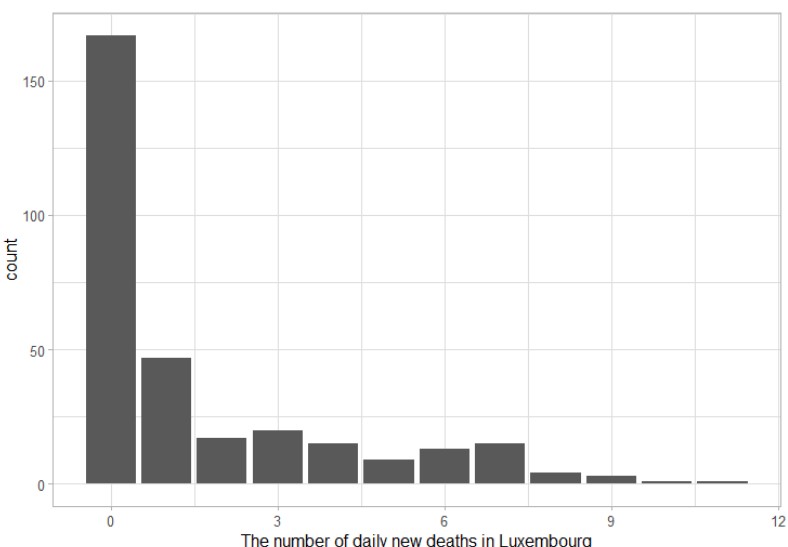

**Figure 1.** The number of daily new deaths in Luxembourg for 2020.

**Table 4.** Descriptive statistics.

| $n$ | Mean | Variance | SD | Skewness | Kurtosis | ID |
|-----|------|----------|-----|----------|----------|-----|
| 312 | 1.6314 | 5.9570 | 2.4407 | 1.5296 | 4.3743 | 3.6514 |

**Table 5.** The log likelihood, AIC, and AICc values of different models.

| Model | ZOICG | ZICG | CG | ZIP | ZIG | ZINB | Poisson | Geometric | NB |
|-------|-------|------|-----|-----|-----|------|---------|-----------|-----|
| $-l$ | 515.186 | 517.120 | 545.206 | 545.900 | 519.611 | 519.200 | 725.088 | 545.206 | 523.379 |
| AIC | 1038.372 | 1040.239 | 1094.413 | 1095.753 | 1043.223 | 1044.302 | 1452.176 | 1092.413 | 1050.758 |
| AICc | 1038.502 | 1040.317 | 1094.452 | 1095.791 | 1043.261 | 1044.379 | 1452.189 | 1092.426 | 1050.797 |

The 95% CIs for the parameters of the ZOICG model obtained by using the five methods are provided in Table 6. The lower and upper bounds for parameters $q_0$ and $q_1$ provided by all of the methods were similar, with the lower bound for the parameter $p$ via the Wald CI being slightly lower and the 95% Wald-based CI for the parameter $\theta$ being much lower than the others.

**Table 6.** Estimation of the number of COVID-19 daily new deaths in Luxembourg in 2020.

| Parameter | Wald | B.uniform | HPD.uniform | B.Jeffreys | HPD.Jeffreys |
|-----------|------|-----------|-------------|------------|--------------|
| $q_0$ | (0.4799,0.5906) 0.1107 | (0.4781,0.5883) 0.1103 | (0.4746,0.5845) 0.1099 | (0.4771,0.5875) 0.1105 | (0.4790, 0.5891) 0.1101 |
| $q_1$ | (0.1110,0.1903) 0.0794 | (0.1148,0.1950) 0.0802 | (0.1148,0.1950) 0.0802 | (0.1136,0.1936) 0.0800 | (0.1137,0.1937) 0.0800 |
| $p$ | (0.7643,0.8610) 0.0967 | (0.8002,0.8636) 0.0634 | (0.8023,0.8651) 0.0628 | (0.8036,0.8658) 0.0621 | (0.8040,0.8658) 0.0618 |
| $\theta$ | (0.0706,0.1004) 0.0298 | (1.4800,1.5060) 0.0260 | (1.4843,1.5061) 0.0218 | (1.4794,1.5043) 0.0249 | (1.4841,1.5047) 0.0207 |

## 6. Discussion

In this paper, a novel four-parameter discrete distribution called the ZOICG distribution is proposed, and its statistical properties are derived. As the model in Equation (6) has a complex pmf, it is more difficult to calculate parameters $\omega_1$ and $\omega_2$ by using the MLEs than by reparameterizing, and so the ZOICG model in Equation (15) is needed. This model requires that parameters $q_0$ and $q_1$ are independent of parameters $p$ and $\theta$,

which means that the proportion of data containing zeros and ones should be treated separately from the rest for the convenience of estimation. From the simulation results, it is clear that all of the methods could detect zeros and ones ($q_0$ and $q_1$) with coverage probabilities close to the nominal level. The MLEs of parameters $p$ and $\theta$ did not have a closed form, so the Newton-type algorithm was applied to solve them numerically. However, the application of this algorithm was not appropriate, especially for parameter $\theta$. Hence, Bayesian analysis was required in this study. However, the derivations of the Bayesian estimates for parameters $p$ and $\theta$ were still complex and difficult to estimate, since their marginal posterior distributions did not have a closed form. Hence, the random walk Metropolis–Hastings steps within a Gibbs sampling algorithm were applied to generate samples for these two parameters.

The simulation results show that the Bayesian methods performed better than the Wald CI based on the MLE. Moreover, the Bayesian method based on the uniform prior was more efficient than the one based on the Jeffreys prior. Since the optimization method is sometimes unsuitable for a model that is complex, the Wald-based CI provided the worst estimates. The efficacy of the proposed ZOICG model for analyzing real data containing excess zeros and ones (new daily COVID-19 deaths in Luxembourg in 2020) was excellent, and so it is recommended in these circumstances. In future research, some characteristics and properties of the ZOICG distribution and statistical methods for estimating the parameters of this model should be further investigated.

## 7. Conclusions

The proposed ZOICG distribution with reparameterization is useful for analyzing datasets with excess zeros and ones. It can detect the proportion of zeros and ones by using five methods that provide CPs close to the nominal level. However, the other parameters of the model ($p$ and $\theta$) are difficult to estimate, since their MLEs and their marginal posterior distributions do not have closed forms. From the simulation results, the Wald CI provided unsuitable CPs for the parameter $\theta$ under all scenarios tested. Hence, the Metropolis–Hastings steps within a Gibbs sampling algorithm were used to estimate them. The results show that the Bayesian method based on the uniform prior performed better than the others in most situations, and so it is recommended for estimating the 95% CIs for the parameters of the ZOICG model.

In the application study, we showed that the proposed ZOICG distribution is appropriate for analyzing datasets with excess zeros and ones, such as the number of new daily COVID-19 deaths in Luxembourg in 2020. Thus, the ZOICG model is suitable for analyzing overdispersed count data, especially those with a high index of dispersion. Furthermore, the proposed model performed well with the Bayesian-derived CI based on a uniform prior in estimating the parameters of the ZOICG model in both the simulation and application studies. Other statistical parameters, such as the mean, will be investigated in future research because they play an important role in statistical inference. There have not been many research articles published on CIs for the mean of a zero-and-one inflated population [25]. Therefore, we will investigate this in the near future.

**Author Contributions:** Conceptualization, S.-A.N. and S.N.; methodology, S.J.; software, S.J.; validation, S.J., S.-A.N., and S.N.; formal analysis, S.J.; investigation, S.J.; resources, S.N.; data curation, S.J.; writing—original draft preparation, S.J.; writing—review and editing, S.-A.N. and S.N.; visualization, S.N.; supervision, S.-A.N.; project administration, S.-A.N.; funding acquisition, S.-A.N. All authors have read and agreed to the published version of the manuscript.

**Funding:** This research received funding support from the National Science, Research and Innovation Fund (NSRF), and King Mongkut's University of Technology North Bangkok (KMUTNB) (Grant No. KMUTNB-FF-65-22).

**Data Availability Statement:** The numbers of COVID-19 new cases and deaths reported per day and per country in the EU and EEA by the European Centre for Disease Prevention and Control (ECDC) are available at https://www.ecdc.europa.eu/en/publications-data/data-daily-new-cases-covid-19 -eueea-country (accessed on 13 September 2022).

**Acknowledgments:** The authors would like to thank the academic editor and the referees for their helpful comments on our manuscripts. The first author acknowledges the generous financial support from the Science Achievement Scholarship of Thailand (SAST).

**Conflicts of Interest:** The authors declare no conflict of interest. The funders had no role in the design of the study; in the collection, analyses, or interpretation of data; in the writing of the manuscript; or in the decision to publish the results.

## Abbreviations

The following abbreviations are used in this manuscript:

| | |
|---|---|
| CI | Confidence interval |
| CP | Coverage probability |
| AL | Average length |
| B.uniform | The dual-tail Bayesian confidence interval with the uniform prior |
| B.Jeffreys | The dual-tail Bayesian confidence interval with the Jeffreys prior |
| HPD.uniform | The highest posterior density interval with the uniform prior |
| HPD.jeffreys | The highest posterior density interval with the Jeffreys prior |
| ZICG | Zero-inflated cosine geometric |
| ZOICG | Zero-and-one inflated cosine geometric |
| ZIP | Zero-inflated Poisson |
| ZIG | Zero-inflated geometric |
| ZINB | Zero-inflated negative binomial |

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
