# Peer review of "A Zero-and-One Inflated Cosine Geometric Distribution and Its Application"

_mathematics, doi:10.3390/math10214012_

Round 1
Reviewer 1 Report
1). Alternative count based models were not discussed in the introduction section and thus, the justification of proposing new model cannot be substantiated.
2). What are the weaknesses of existing count models and how does the proposed one is an improvement?
3). The Bayesian approach used by the authors make the paper very unique and will most likely generates citations.
4). The authors should relate the discussion with related count models.
5). The authors should verify that the equations are void of errors.
6). Is it unclear whether the results is reproducible using the same data.
7). Establish that the MLE may yield results that has large variance or efficiency.
Author Response
Journal: Mathematics
Title name: A Zero-and-One Inflated Cosine Geometric Distribution and its Application
Authors: Sunisa Junnumtuam, Sa-Aat Niwitpong and Suparat Niwitpong
Dear reviewer,
We are very grateful for your comments to our manuscript. We revised the manuscript in accordance with your advice. Here below is one-by-one response to your comments.
Lists of correction.
Reviewer 1:
1). Alternative count based models were not discussed in the introduction section and thus, the justification of proposing new model cannot be substantiated.
Response: In Section Introduction, paragraph 3, we add more research about alternative zero-and-one inflated count model as follows:
“However, overdispersion arises when large numbers of zeros and ones occur simultaneously. Since the ZI model is not an appropriate for model such data, it has been extended to the zero-and-one-inflated (ZOI) model combined with several other distributions to produce the ZOI Poisson (ZOIP), ZOI geometric, ZOI negative binomial-beta exponential (ZOINB-BE), and ZOIP-Lindley mixed distributions. Zhang et al. [8] explored the ZOIP distribution extended from the ZIP distribution [9]; five equivalent stochastic representations for a ZOIP random variable were presented and their important distributional properties were derived. Tang et al. [10] provided a ZOIP model and analyzed two real datasets of Legionnaires’ disease cases in Singapore and accidental deaths in Detroit, USA, both of which contain a high proportion of zeros and ones. They used the data augmentation method to obtain maximum likelihood estimators (MLEs) via the EM algorithm and Bayesian estimations via Gibbs sampling. By obtaining the lowest Akaike information criterion (AIC), deviance information criterion (DIC), and widely applicable information criterion (WAIC) values for the ZOIP, they reported that it was more appropriate for these datasets than the ZIP model. Liu et al. [11] analyzed two real datasets of Legionnaires’ disease cases in Singapore and accidental deaths in Detroit, USA, by using the ZOIP model with both maximum likelihood and Bayesian estimation; they found that the ZOIP model was more appropriate than the ZIP model for analyzing these two real datasets. Liu et al. [12] studied the number of daily accidental deaths 1994 available from the NMMAPS database by using a ZOIP regression model and investigated maximum likelihood and Bayesian estimation, the expectation-maximization algorithm, the generalized expectation-maximization algorithm, and Gibbs’ sampling to estimate its parameters. They found that the ZOIP regression model was more appropriate than the ZIP model for analyzing this accidental deaths dataset as it attained lower AIC, Bayesian information criterion (BIC), DIC, and WAIC values. Xiao et al. [13] constructed a ZOI geometric distribution regression model and introduced Pólya-gamma latent variables in the Bayesian inference; they found that it was more suitable for analyzing a doctoral dissertation dataset than the ZOIP regression model. Jornsatian and Bodhisuwan [14] presented a ZOI negative binomial-beta exponential (ZOINB-BE) distribution and investigated some of its important properties, and analyzed three real datasets: the number of visits to the doctor in Germany in 1998 (the COUNT package in the R programming suite), the number of accidental injuries in the US in 2001 [15-16], and the number of monthly crimes in Greece from 1982–1993 [17-18] and found that the ZOINB-BE distribution was the most appropriate to fit these data by attaining the lowest log-likelihood, AIC, mean absolute error, and root-mean-squared error values. Tajuddin et al. [19] introduced the ZOIP-Lindley distribution and developed MLEs and method-of-moment estimators for its parameters; they found that it was the most appropriate for analyzing two datasets: the number of criminal acts [20] and the number of stillbirths of New Zealand white rabbits [21]. Mohammadi et el. [22] introduced a zero-and-one inflated INAR(1) process with Poisson-Lindley distribution and analyzed the number of abortions of animals reported monthly which contains large proportions of zeros and ones; they found that the proposed model has the best fit based on the AIC, BIC, log-likelihood and root-mean-square differences between observations and predictions (RMS) criteria.”
2). What are the weaknesses of existing count models and how does the proposed one is an improvement?
Response: Even though there are some zero-and-one inflated count models that were used to analyze count data with excess zeros and ones, the zero-and-one inflated cosine geometric model is proposed to be an alternative model. Since the CG distribution is flexible and appropriate for over-dispersed count data (Chesneau [23]), this characteristic shows that the zero-and-one inflated cosine geometric is suitable for over-dispersed count data with excess zeros and ones. As the empirical result shows that the ZICG model is the best fit for the number of new COVID-19 deaths per day in Luxembourg from 24 February 2020 to 31 December 2020, which contains a large proportion of zero and one and a high index of dispersion (ID = 3.6514). However, the ZOICG model has four parameters that are difficult to estimate since they are not closed forms for the MLEs. After re-parameterizing, the proportions of zero and one are easier to estimate, but the rest of the parameters are still not in closed form. Hence, the Newton-type algorithm in the R program is applied to the MLEs and the Metropolis-Hasting steps within Gibbs sampling are used to generate the sample to estimate the Bayesian estimator, which consumes time for optimization, and sometimes the optimization does not provide the best estimate parameters that lead to poor estimation, such as for parameter from the Wald CI.
3). The Bayesian approach used by the authors make the paper very unique and will most likely generates citations.
Response: Thank you so much for your compliment.
4). The authors should relate the discussion with related count models.
Response: The first paragraph of the section discussion, we add more details as follows:
“In this paper, a novel four-parameter discrete distribution called the ZOICG distribution is proposed and its statistical properties are derived. As the model in Equation (6) has a complex pmf, it is more difficult to calculate parameters and by using the MLEs than by re-parameterizing, and so the ZOICG model in Equation (15) is needed. This model requires that parameters and are independent of parameters and , which means that the proportion of data containing zeros and ones should be treated separately from the rest for the convenience of estimation. From the simulation results, it is clear that all of the methods can detect zeros and ones ( and ) with coverage probabilities close to the nominal level. The MLEs of parameters and do not have a closed form, so the Newton-type algorithm was applied to solve them numerically. However, the application of this algorithm is not appropriate, especially for parameter . Hence Bayesian analysis was required in this study. However, the derivations of Bayesian estimates for parameters and are still complex and difficult to estimate since their marginal posterior distributions do not have a closed form. Hence, the random-walk Metropolis-Hastings steps within a Gibbs sampling algorithm were applied to generate samples for these two parameters.”
5). The authors should verify that the equations are void of errors.
Response: The equations are derived and calculated in the R program which is reliable.
6). Is it unclear whether the results is reproducible using the same data.
Response: In a simulation study, a function set.seed() was used to set the same data set for Monte Carlo simulation study, which confirms that the replications for the simulation study using the same data.
7). Establish that the MLE may yield results that has large variance or efficiency.
Response: Since the MLE for parameter p or has no closed form, the Newton-type algorithm is applied to optimize the likelihood function. As a result, the Wald CI for parameter has a low CP because sometimes the iterative did not provide the best optimization which led to poor estimation.
Best Regards,
The authors

Reviewer 2 Report
Let's start by noting that "A Zeros-and-Ones Inflated Cosine Geometric Distribution and its Application" is a text with very poor punctuation and comma, to the point of complicating its understanding and being able to jeopardize the respective evaluation. For now, a careful review of the text by the authors is recommended, which only they can do because they certainly know what they mean.
As far as I could see the authors start from the principle, see "abstract", that (ZOICG) distribution is suitable for analyzing count data containing both excess zeros and ones occurring", which gives the idea that this distribution already exists and has these qualities. Later, in the Introduction, they say that they are going to create this distribution. Therefore, it does not yet exist and will be created. It is necessary to clarify this situation and say why you think the distribution you are going to create has these properties ( the literature review that they present does not clarify it at all) In short: They must clearly state the motivation for creating this distribution.
There follows a worthwhile job of determining the parameters and confidence intervals, and other quantities of statistical interest, using an appreciable amount of estimation methods.
Simulations and application to a real case serve to illustrate and compare the performance of different estimation methods, and the performance of the model in data analysis.
The conclusions are too short, focused only on the performance of the estimation methods, and completely failing to critically analyze the model's qualities and defects.
Author Response
Journal: Mathematics
Title name: A Zero-and-One Inflated Cosine Geometric Distribution and its Application
Authors: Sunisa Junnumtuam, Sa-Aat Niwitpong and Suparat Niwitpong
Dear reviewer,
We are very grateful for your comments to our manuscript. We revised the manuscript in accordance with your advice. Here below is one-by-one response to your comments.
Reviewer 2:
- Let's start by noting that "A Zeros-and-Ones Inflated Cosine Geometric Distribution and its Application" is a text with very poor punctuation and comma, to the point of complicating its understanding and being able to jeopardize the respective evaluation. For now, a careful review of the text by the authors is recommended, which only they can do because they certainly know what they mean.
Response: The punctuation and comma were revised and checked by a native English-speaking proofreader.
- As far as I could see the authors start from the principle, see "abstract", that (ZOICG) distribution is suitable for analyzing count data containing both excess zeros and ones occurring", which gives the idea that this distribution already exists and has these qualities. Later, in the Introduction, they say that they are going to create this distribution. Therefore, it does not yet exist and will be created. It is necessary to clarify this situation and say why you think the distribution you are going to create has these properties (the literature review that they present does not clarify it at all) In short: They must clearly state the motivation for creating this distribution.
Response: Section Introduction, we add some remarkable motivation as follows.
“Over-dispersed count data with excess zeros frequently occur in various fields such as natural science (e.g., the number of torrential rainfall incidences at the Daegu and the Busan rain gauge stations in South Korea [1]), medical science (e.g., the DMFT (decayed, missing, and filled teeth) index in dentistry [2], the number of falls by people with Parkinson’s disease [3], and the number of daily COVID-19 deaths in Thailand [4]), and insurance (e.g., the frequency of health insurance claims [5]). Although the Poisson model is widely used to analyze such discrete data, the strong assumption of the equality of the mean and variance implies that it is inadequate for modeling data where the variance is larger than the mean, which is termed "overdispersion" [6] and can arise in various ways.
One common source of overdispersion is when the observed counts contain excess zeros, which has motivated the creation of modified count models such as zero-inflated (ZI) and hurdle. These two classes can be considered as finite mixture models of two subpopulations (components): (1) the observations contain zeros with probability and (2) the observations occur with probability 1- from the baseline distribution of a ZI model and the zero-truncated probability mass function (pmf) (the non-zero part) of a hurdle model. One of the most popular ZI models is the ZI Poisson (ZIP) model where the Poisson distribution is the baseline distribution [7].
However, overdispersion arises when large numbers of zeros and ones occur simultaneously. Since the ZI model is not an appropriate for model such data, it has been extended to the zero-and-one-inflated (ZOI) model combined with several other distributions to produce the ZOI Poisson (ZOIP), ZOI geometric, ZOI negative binomial-beta exponential (ZOINB-BE), and ZOIP-Lindley mixed distributions. Zhang et al. [8] explored the ZOIP distribution extended from the ZIP distribution [9]; five equivalent stochastic representations for a ZOIP random variable were presented and their important distributional properties were derived. Tang et al. [10] provided a ZOIP model and analyzed two real datasets of Legionnaires’ disease cases in Singapore and accidental deaths in Detroit, USA, both of which contain a high proportion of zeros and ones. They used the data augmentation method to obtain maximum likelihood estimators (MLEs) via the EM algorithm and Bayesian estimations via Gibbs sampling. By obtaining the lowest Akaike information criterion (AIC), deviance information criterion (DIC), and widely applicable information criterion (WAIC) values for the ZOIP, they reported that it was more appropriate for these datasets than the ZIP model. Liu et al. [11] analyzed two real datasets of Legionnaires’ disease cases in Singapore and accidental deaths in Detroit, USA, by using the ZOIP model with both maximum likelihood and Bayesian estimation; they found that the ZOIP model was more appropriate than the ZIP model for analyzing these two real datasets. Liu et al. [12] studied the number of daily accidental deaths 1994 available from the NMMAPS database by using a ZOIP regression model and investigated maximum likelihood and Bayesian estimation, the expectation-maximization algorithm, the generalized expectation-maximization algorithm, and Gibbs’ sampling to estimate its parameters. They found that the ZOIP regression model was more appropriate than the ZIP model for analyzing this accidental deaths dataset as it attained lower AIC, Bayesian information criterion (BIC), DIC, and WAIC values. Xiao et al. [13] constructed a ZOI geometric distribution regression model and introduced Pólya-gamma latent variables in the Bayesian inference; they found that it was more suitable for analyzing a doctoral dissertation dataset than the ZOIP regression model. Jornsatian and Bodhisuwan [14] presented a ZOI negative binomial-beta exponential (ZOINB-BE) distribution and investigated some of its important properties, and analyzed three real datasets: the number of visits to the doctor in Germany in 1998 (the COUNT package in the R programming suite), the number of accidental injuries in the US in 2001 [15-16], and the number of monthly crimes in Greece from 1982–1993 [17-18] and found that the ZOINB-BE distribution was the most appropriate to fit these data by attaining the lowest log-likelihood, AIC, mean absolute error, and root-mean-squared error values. Tajuddin et al. [19] introduced the ZOIP-Lindley distribution and developed MLEs and method-of-moment estimators for its parameters; they found that it was the most appropriate for analyzing two data-sets: the number of criminal acts [20] and the number of stillbirths of New Zealand white rabbits [21]. Mohammadi et el. [22] introduced a zero-and-one inflated INAR(1) process with Poisson-Lindley distribution and analyzed the number of abortions of animals reported monthly which contains large proportions of zeros and ones; they found that the proposed model has the best fit based on the AIC, BIC, log-likelihood and root-mean-square differences between observations and predictions (RMS) criteria.
Besides these mixed distributions, another one is the two-parameter discrete cosine geometric (CG) distribution which is proposed by Chesneau [23]. This belongs to the family of weighted geometric distributions with its pmf given by
, (1)
where and
. (2)
If , then we can obtain and is a standard geometric distribution. A weighted geometric distribution makes the CG distribution more flexible than the standard geometric distribution. The former is better for analyzing over-dispersed data than the Poisson, geometric, negative binomial (NB), and weighted NB distributions. Junnumtuam et al. [24] extended it and proposed the ZICG distribution (with CG as the baseline distribution) and reported that it is appropriate for fitting over-dispersed count data containing excess zeros, such as the number of daily COVID-19-positive cases at the Tokyo 2020 Olympic Games.
Since studies on ZOI count data have gained much research interest, and the CG distribution is appropriate with over-dispersed data, we can extend it to the novel four-parameter discrete ZOICG distribution and derive some of its statistical properties such as the pmf, moment generating function (mgf), mean, variance, and Fisher information. Moreover, its parameters are estimated by deriving their confidence intervals (CIs). There are several examples of applying CIs to analyze ZI and ZOI count data. Liu et al. [11] considered both point and interval estimation for the parameters of a ZOIP model and compared Bayesian estimation using either the Jeffreys’ or reference prior with the MLE method via Monte Carlo simulation; the results indicate that the Bayesian estimates performed slightly better when the sample size was small or moderate. Tian et el. [25] proposed CIs for the mean of a zero-and-one inflated population by using the jackknife empirical likelihood and adjusted jackknife empirical likelihood methods. Wald CIs were constructed for the parameters in the Bernoulli component of ZIP and hurdle models by [26] and for the ZIP mean by [27].
This motivated us to study the confidence intervals for parameters of the ZOICG distribution to estimate the over-dispersed data which contains a large proportion of zero and one with a high index of dispersion by using five methods: a Wald CI based on the MLE, equal-tailed Bayesian CIs based on the uniform or Jeffreys’ prior, and highest posterior density (HPD) intervals based on the uniform or Jeffreys’ prior. Furthermore, real data containing excess zeros and ones (the number of new daily COVID-19 deaths in Luxembourg in 2020) were used to investigate their efficacies.”
- There follows a worthwhile job of determining the parameters and confidence intervals, and other quantities of statistical interest, using an appreciable amount of estimation methods.
Simulations and application to a real case serve to illustrate and compare the performance of different estimation methods, and the performance of the model in data analysis.
The conclusions are too short, focused only on the performance of the estimation methods, and completely failing to critically analyze the model's qualities and defects.
Response: In the conclusions, we add more detail as follows:
“The proposed ZOICG distribution with re-parameterization is useful for analyzing datasets with excess zeros and ones. It can detect the proportion of zeros and ones by using five methods that provided CPs close to the nominal level. However, the other parameters of the model ( and ) are difficult to estimate since their MLEs and their marginal posterior distributions do not have a closed form. From the simulation results, the Wald CI provided unsuitable CPs for parameter under all scenarios tested. Hence, the Metropolis-Hastings steps within a Gibbs sampling algorithm were used to estimate them. The results show that the Bayesian method based on the uniform prior performed better than the others in most situations, and so it is recommended for estimating the 95% CIs for the parameters of the ZOICG model.
In the application study, we showed that the proposed ZOICG distribution is appropriate for analyzing datasets with excess zeros and ones, such as the number of new daily COVID-19 deaths in Luxembourg in 2020. Thus, the ZOICG model is suitable for analyzing over-dispersed count data, especially with a high index of dispersion. Furthermore, the proposed model performed well with the Bayesian-derived CI based on uniform prior to estimate the parameters of the ZOICG model in both the simulation and application studies. Other statistical parameters, such as the mean, will be investigated in future research because they play an important role in statistical inference. There have not been many research articles published on CIs for the mean of a zero-and-one inflated population [25], and so we will investigate it in the near future.”
Best Regards,
The authors

Round 2
Reviewer 2 Report
The authors satisfactorily answered the questions raised, having corrected and expanded the text in order to meet them,
This resulted in a text of sufficient quality to merit publication in Mathematics.